# An acid-tolerance response system protecting exponentially growing *Escherichia coli*

Ying Xu [1,2,8], Zhe Zhao [1,2,8], Wenhua Tong [1,7], Yamei Ding [3], Bin Liu[4], Yixin Shi[5], Jichao Wang [1], Shenmei Sun[1,2], Min Liu [1], Yuhui Wang[4], Qingsheng Qi[6], Mo Xian[1✉] & Guang Zhao [1,6✉]

The ability to grow at moderate acidic conditions (pH 4.0–5.0) is important to *Escherichia coli* colonization of the host's intestine. Several regulatory systems are known to control acid resistance in *E. coli*, enabling the bacteria to survive under acidic conditions without growth. Here, we characterize an acid-tolerance response (ATR) system and its regulatory circuit, required for *E. coli* exponential growth at pH 4.2. A two-component system CpxRA directly senses acidification through protonation of CpxA periplasmic histidine residues, and upregulates the *fabA* and *fabB* genes, leading to increased production of unsaturated fatty acids. Changes in lipid composition decrease membrane fluidity, $F_OF_1$-ATPase activity, and improve intracellular pH homeostasis. The ATR system is important for *E. coli* survival in the mouse intestine and for production of higher level of 3-hydroxypropionate during fermentation. Furthermore, this ATR system appears to be conserved in other Gram-negative bacteria.

[1] CAS Key Lab of Biobased Materials, Qingdao Institute of Bioenergy and Bioprocess Technology, Chinese Academy of Sciences, 266101 Qingdao, China. [2] University of Chinese Academy of Sciences, 100049 Beijing, China. [3] Institute of Oceanology, Chinese Academy of Sciences, 266071 Qingdao, China. [4] TEDA Institute of Biological Sciences and Biotechnology, Nankai University, TEDA, 300457 Tianjin, China. [5] School of Life Sciences, Arizona State University, Tempe, AZ 85281, USA. [6] State Key Laboratory of Microbial Technology, Shandong University, 266237 Qingdao, China. [7] Present address: Sichuan University of Science and Engineering, 644000 Yibin, Sichuan, China. [8] These authors contributed equally: Ying Xu, Zhe Zhao. ✉email: xianmo@qibebt.ac.cn; zhaoguang@qibebt.ac.cn

Enteric bacteria such as *Escherichia coli* (*E. coli*) and *Salmonella* can colonize and cause disease in the human intestinal tract. They have to combat acidic environments during the whole process of invading the host. With pH values as low as 1.5–2.5, the stomach has been recognized as a natural antibiotic barrier[1]. With their passage into the small intestine, *E. coli* cells will encounter a less acidic environment (pH 4.0–6.0) with the presence of organic acids produced by the normal intestinal flora[2].

*E. coli* has developed variable acidic stress response systems, including the acid resistance (AR) systems response to extreme acid stress and the acid tolerance response (ATR) system towards mild and moderate acid stress[3,4]. Up to now, five AR systems, AR1−AR5, are reported. The AR1 system is activated by alternative σ factor (RpoS) and cAMP receptor protein (CRP)[5,6]. Due to the involvement of CRP, the AR1 system is repressed by glucose. The AR2−AR5 systems are all dependent on a specific extracellular amino acid, and consist of an antiporter as well as a decarboxylase enzyme that is usually induced by low pH and extracellular amino acid[3,7], except that AR2 can be induced at acidic pH in the absence of glutamate[8]. They confer acid resistance by consumption of intracellular protons in amino acid decarboxylation reaction to produce a less acidic internal pH, using glutamate, arginine, lysine and ornithine as their corresponding substrates, respectively[1,5,9–12]. All five AR systems can protect stationary phase cells from the extreme acidity and prolong survival, while only AR2 and AR3 were reported to function during the exponential phase[5,13].

Among AR systems, AR2 is by far the most effective and the most complex. The glutamate decarboxylase isoforms, GadA and GadB, and the glutamate/γ-aminobutyric acid antiporter GadC are key components of AR2, and their regulation relies on the action of over 20 proteins and 3 small noncoding RNAs, including two-component systems EvgAS and PhoPQ; regulatory proteins RpoS, GadE, RcsB, GadX, GadW and HNS; protease ClpXP and Lon; and small RNAs DsrA, GadY and GcvB, which together form a regulatory network with high level of complexity (for a review, see refs. [3,7]). The periplasmic chaperons HdeAB and their cytoplasmic counterpart Hsp31, which assist the refolding of denatured proteins during the acid stress[7,14,15], are also induced as part of the AR2 regulon[16,17].

The ATR system, though poorly understood, is induced by exposing *E. coli* cells to moderate acid stress (pH 4.5–5.8), and will protect cells from a subsequent challenge of extreme acid pH (pH 2.0–3.0)[4,6]. ATR can be activated during adaptation at mild acidic pH by the regulators Fur and PhoPQ in exponential phase cells and by RpoS and OmpR in stationary phase cells, but the stationary phase cells are much more tolerant to acid than the log phase cells[3,4].

Benefited from the complicated AR and ATR systems, *E. coli* can survive without growth for several hours at pH 2.0[1,18–20], and the acid limit for growth of *E. coli* is pH 4.0 in rich medium, or pH 4.5 in minimal medium[6,18,20–22]. So, *E. coli* will experience the transition of pH from no-growth to growth conditions when passing through the stomach and entering the intestine. It is exceptionally important to elucidate how *E. coli* adapts to and grows at pH 4.0–5.0, because the capability of bacteria to outgrow hundreds of competing species in gut microbiome in this lower range of growth pH will determine which strain can colonize the gut[18]. Unfortunately, we still barely know that.

In this study, we challenged the exponentially growing cells of *E. coli* at pH 4.2, and characterized a regulatory circuit required for bacterial growth under moderate acidic conditions through modulation of the membrane lipid composition. The two-component system CpxRA directly senses acidification through protonation of the CpxA periplasmic histidine residues, and thus activates transcription of the essential genes *fabA* and *fabB* in biosynthesis of unsaturated fatty acids (UFAs) to enhance the UFAs content in membrane lipid. This mechanism enables *E. coli* to grow at acidic pH, and also functions in diverse bacterial species.

## Results

**UFAs are required for *E. coli* growth under acidic pH**. We carried out a screening to characterize an *E. coli* ATR system required for bacterial survival in exponential growth. At first, the overnight culture of *E. coli* BW25113 wild-type strain was directly inoculated into minimal medium E at pH 4.2 using glucose as sole carbon source without supplement of any amino acid. However, the cells were rapidly killed, even preadapted at pH 5.0 (Supplementary Fig. 1). Then the BW25113 strain was grown in medium E at pH 7.0 to a cell density of $\approx 3 \times 10^8$ CFU mL$^{-1}$, and the cells were collected, washed and transferred into the same medium at pH 7.0 and 4.2, respectively. While the cells at pH 7.0 grew normally, the cell density at pH 4.2 decreased continuously, which led to reduction of the ratio of CFU at pH 4.2 vs. pH 7.0 to $0.21 \pm 0.02$ after 1 h of exposure at different pH (Fig. 1a). This result confirmed that exponentially growing *E. coli* was susceptible to this moderate acidic condition.

In previous studies, survival was used to measure the bacterial acid resistance, which is calculated by the formula, survival (%) = (CFU after acid challenge/CFU before acid challenge) × 100%. However, survival was determined under conditions *E. coli* can only survive without growth, and is not suitable here as we are trying to figure out how *E. coli* grows at moderate acidic pH. So, the CFU ratio (pH 4.2/pH 7.0) is calculated to represent the acid tolerance of exponentially growing *E. coli* because growth of *E. coli* at pH 7.0 is steady. A value of CFU ratio close to 1 indicates that *E. coli* grows at pH 4.2 to a cell density similar to that at pH 7.0.

Phospholipids were extracted from exponential phase cells after exposure of 1 h to pH 7.0 and 4.2, and analyzed as previously described[23]. The results showed that exposure to such acidic condition caused a change of membrane lipid composition in *E. coli* cells (Fig. 1b). Specifically, levels of the UFAs, including palmitoleic acid (C16:1) and oleic acid (C18:1), were elevated by 3.83- and 1.66-fold; meanwhile, the levels of palmitic acid (C16:0) and stearic acid (C18:0) were reduced. These five fatty acids shown in Fig. 1b represented more than 96% of total fatty acids in *E. coli* cells, consistent with those results reported previously[24,25]. As a consequence, the ratio of unsaturated to saturated fatty acids increased from 0.11 at pH 7.0 to 0.32 at pH 4.2. Comparable to this result in *E. coli*, a shift in the unsaturated/saturated ratio in response to acid stress was also observed in *Streptococcus mutants*[26,27].

We also found that the expression of two essential genes required for UFAs biosynthesis, *fabA* and *fabB* (Fig. 1c), was significantly upregulated under the acidic condition since their protein and mRNA levels were elevated (Fig. 1d, e). A previous study showed that overexpression of *fabA* and *fabB* increased UFA contents in *E. coli*[28]. Therefore, it is plausible that overexpression of *fabA* and *fabB* may allow *E. coli* to grow under pH 4.2. To test this hypothesis, we cloned *fabA* and *fabB* into vector pTrcHis2B, and introduced them into BW25113 wild-type strain. The strain carrying empty vector presented a CFU ratio of $0.26 \pm 0.03$, similar with that of BW25113 wild-type strain, while either strain with *fabA* or *fabB* showed much higher acid tolerance at pH 4.2 (Fig. 1f).

In agreement with this result, strains harboring a temperature-sensitive mutant of *fabA* or *fabB* gene became much more susceptible to acid at 42 °C, whereas the wild-type strain showed

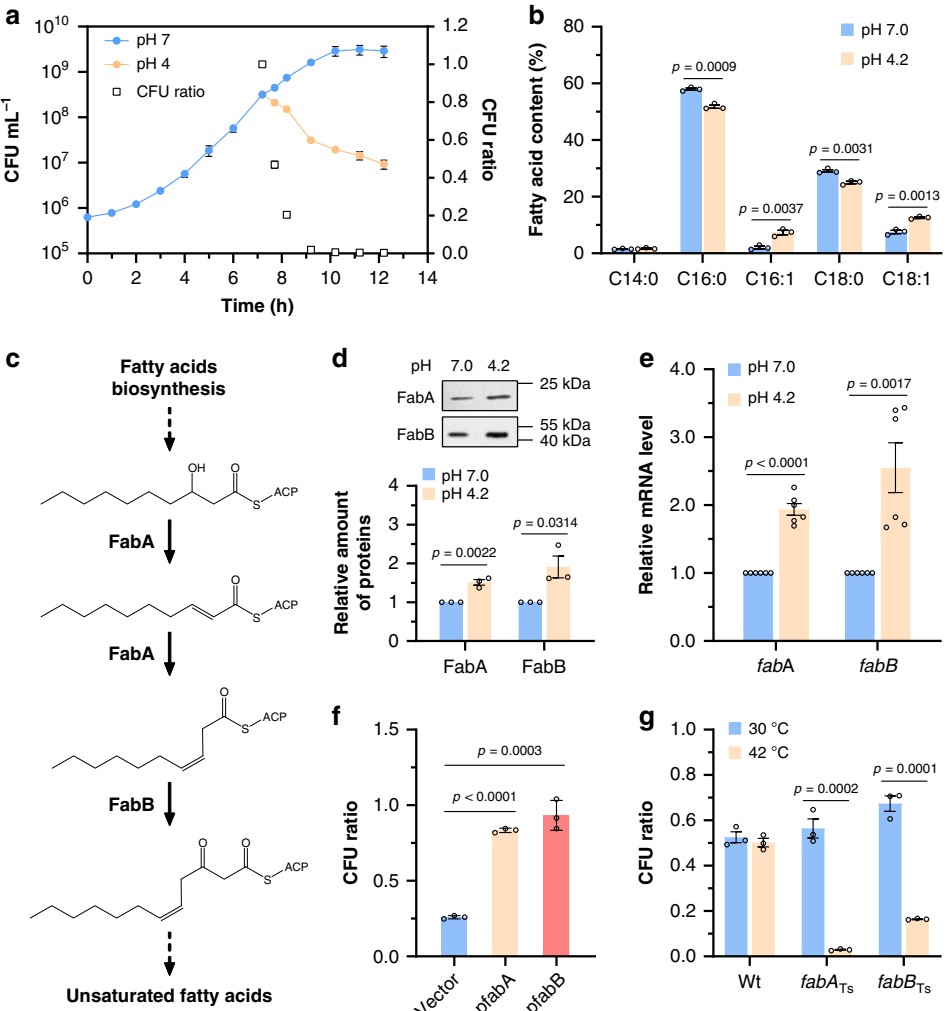

**Fig. 1 Improved acid tolerance of exponentially growing *E. coli* caused by increased production of unsaturated fatty acids. a** Growth of *E. coli* BW25113 strain at different pH. The strain was grown to $3 \times 10^8$ CFU mL$^{-1}$ in minimal medium at pH 7.0, and transferred into the same medium at pH 7.0 and pH 4.2 ($n = 3$ biologically independent samples). CFU ratio = CFU at pH 4.2/CFU at pH 7.0. **b** Membrane lipid composition of BW25113 strain after acidic challenge at pH 4.2 for 1 h. The compositions were determined by GC-MS, and fatty acid content is given as the relative peak area [(peak area of one fatty acid/total peak area) × 100%] ($n = 3$ biologically independent samples). **c** Biosynthetic pathway of unsaturated fatty acids, in which FabA and FabB, 3-hydroxyacyl-ACP dehydratase/isomerase and β-ketoacyl-ACP synthase play essential roles. **d** FabA-His$_6$ and FabB-His$_6$ protein level of BW25113 strain after 1 h of exposure to pH 7.0 and 4.2 determined by western blot. The relative amount of each protein was determined using ImageJ ($n = 3$ biologically independent samples). **e** Relative mRNA level of *fabA* and *fabB* in BW25113 strain after 1 h of exposure to pH 7.0 and 4.2 determined by qRT-PCR ($n = 2$ biologically independent samples with three technical repeats). **f** Tolerance of BW25113 strain carrying empty vector pTrcHis2B, p*fabA* or p*fabB*, respectively, after acidic challenge at pH 4.2 for 1 h ($n = 3$ biologically independent samples). **g** Tolerance of BW25113 strain, temperature-sensitive FabA and FabB mutants after acidic challenge at pH 4.2 at 30 or 42 °C for 0.5 h ($n = 3$ biologically independent samples). Error bars, mean ± standard error of mean (SEM). Two-tailed Student's *t* tests were performed to determine the statistical significance for two group comparisons. The source data are provided as a Source Data file.

similar acid tolerance at different temperatures (Fig. 1g). Taken together, these results demonstrated that elevating the biosynthesis of UFAs enhanced the acid tolerance of exponentially growing *E. coli*.

**Transcription of *fabA* and *fabB* is activated by CpxRA system.** To find out the possible regulator of these loci, we compared the DNA sequences of *fabA* and *fabB* promotor regions, and found a conserved sequence GTAAA-(5 nt)-GCAAA (Fig. 2a, b), which was similar to the identified CpxR recognition site[29]. This raised the hypothesis that the transcription of *fabA* and *fabB* is regulated by two-component system CpxRA, which consists of a sensor histidine kinase CpxA and a cytoplasmic response regulator

CpxR. Since overexpression of outer membrane protein NlpE was identified as a Cpx-specific activation signal[30,31], we cloned the *nlpE* gene and tested its effect on expression of *fabA* and *fabB* loci. As shown in Fig. 2c, d, excessive NlpE protein significantly enhanced the expression level of *fabA* and *fabB* at pH 7.0, and deletion of *cpxR* totally abolished this activating effect of NlpE. Besides those, NlpE overexpression also raised the UFAs content in *E. coli* membrane lipid at pH 7.0 (Fig. 2e) and *E. coli* acid tolerance at pH 4.2 (Fig. 2f) in a CpxR-dependent manner since *cpxR* knockout mutant displayed higher susceptibility to acidic challenge than wild-type strain even overexpressing NlpE (Fig. 2f). These observations collectively demonstrate that the CpxRA system activates transcription of *fabA* and *fabB*.

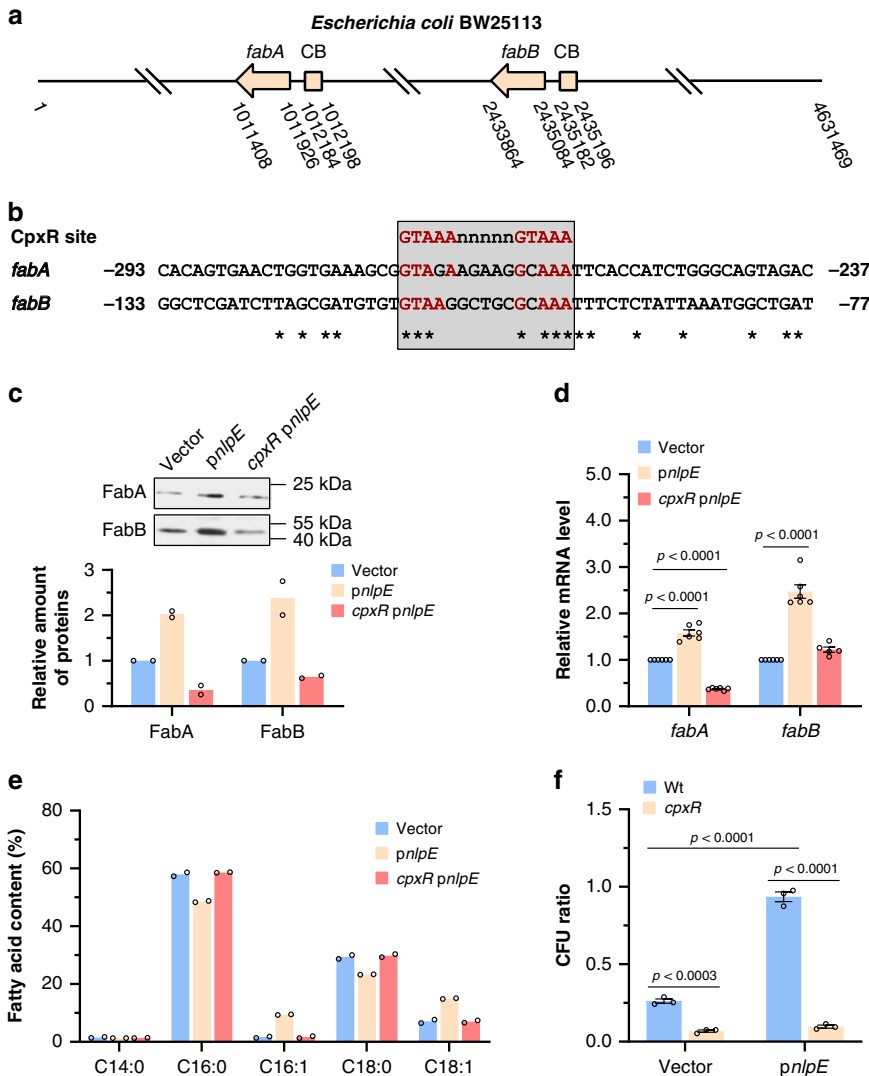

**Fig. 2 Transcription of *fabA* and *fabB* is activated by two-component system CpxRA. a** Schematic diagram showing the genomic locations of *fabA and fabB* genes and corresponding CpxR boxes. CB CpxR box. Arrows show the transcription direction of *fabA* and *fabB* genes. **b** Alignment of CpxR binding sequence and the promoter sequences of *fabA* and *fabB* genes. The conserved CpxR binding sequence was highlighted in red. **c** Western blot analysis of FabA-His$_6$ and FabB-His$_6$ protein in BW25113 strains carrying empty vector or p*nlpE*, and *cpxR* mutant carrying p*nlpE* grown at pH 7.0. Overexpression of outer membrane protein NlpE was reported as a specific activating signal of CpxRA system. The relative amount of each protein was determined using ImageJ from two independent experiments. **d** Relative mRNA level of *fabA* and *fabB* in BW25113 strains carrying empty vector or p*nlpE*, and *cpxR* mutant carrying p*nlpE* grown at pH 7.0 determined by qRT-PCR ($n = 2$ biologically independent samples with three technical repeats). **e** Membrane lipid composition of BW25113 strains carrying empty vector or p*nlpE*, and *cpxR* mutant carrying p*nlpE* grown at pH 7.0 ($n = 2$ biologically independent samples). The compositions were determined by GC-MS, and fatty acid content is given as the relative peak area [(peak area of one fatty acid/total peak area) × 100%]. **f** Acid tolerance of BW25113 strains carrying empty vector or p*nlpE*, and *cpxR* mutant carrying vector or p*nlpE* after acidic challenge at pH 4.2 for 1 h ($n = 3$ biologically independent samples). Error bars, mean ± SEM. Two-tailed Student's *t* tests were performed to determine the statistical significance for two group comparisons. The source data are provided as a Source Data file.

**Transcription start site of *fab* genes controlled by CpxRA.** We mapped the transcription starts of *fabA* and *fabB* using RACE (rapid-amplification of cDNA ends) experiment. As was previously reported, for *fabA* gene, two transcription initiation sites were detected under growth at pH 7.0: S1 positively regulated by FadR[32] and S2 negatively regulated by FabR[33]. Interestingly, when the cell growth was at pH 4.2, our results indicated that the *fabA* gene has a third transcription initiation site located at the 203 bp upstream of the start codon, the S3 (summarized in Fig. 3a). For *fabB* gene, the transcription was initiated 37 bp upstream of the start codon in both conditions (summarized in Fig. 3b), consistent with previous reports[34].

qRT-PCR result showed that the mRNA level of *fabA* and *fabB* was upregulated by NlpE overexpression in a *fabR fadR* double mutant strain, indicating that they had no obvious effect on *fabA* and *fabB* gene expression under CpxRA-dependent activation (Supplementary Fig. 2).

**CpxR protein directly binds to the *fabA* and *fabB* promoters.** To verify the putative CpxR recognition sites, two mutant strains were constructed by site-directed mutagenesis, in which the CpxR binding box on *fabA or fabB* promoter region was replaced by CATCT-(5 nt)-CATCT sequence, and expression of *fabA* and *fabB* was determined. Both western blot and qRT-PCR results

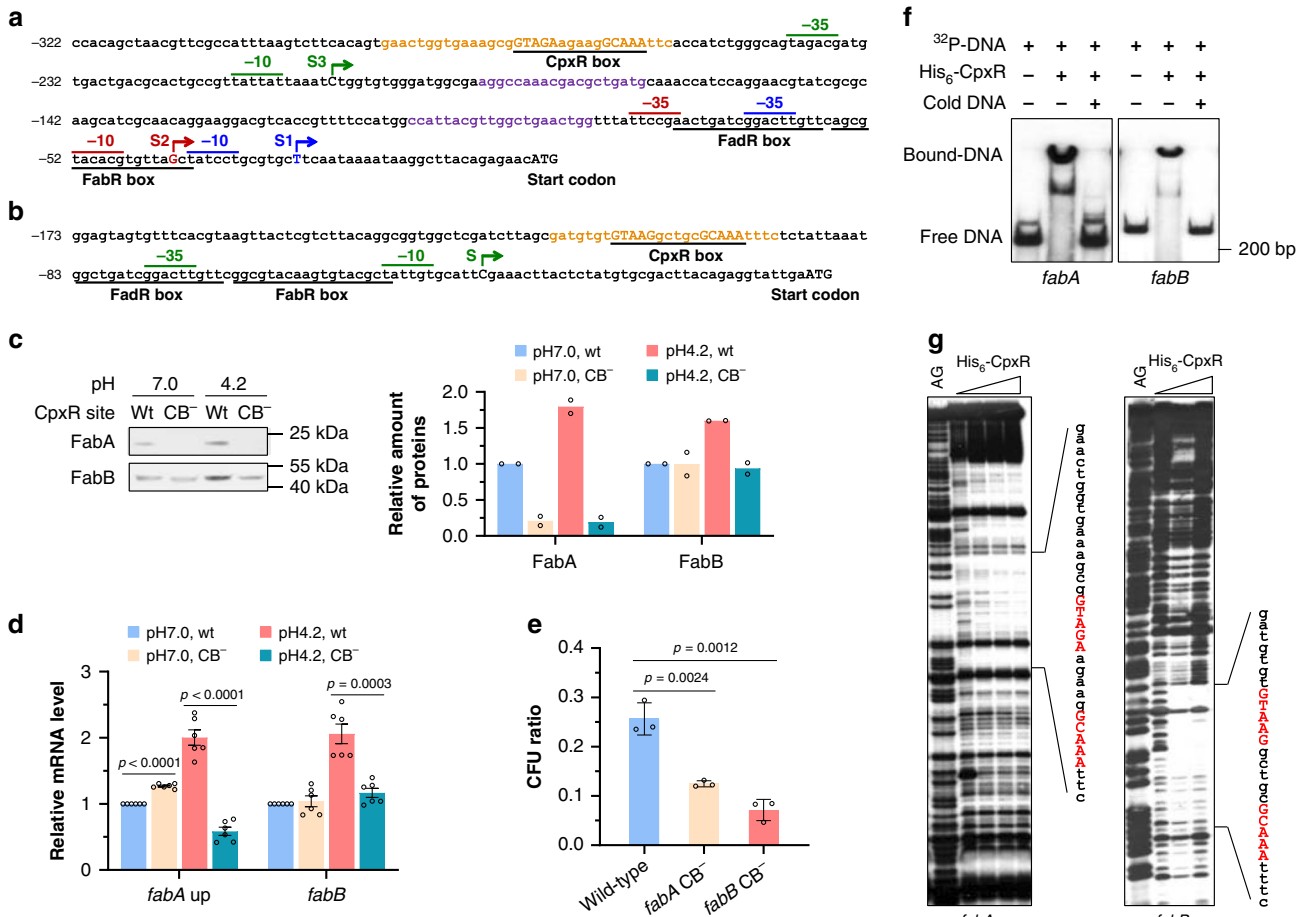

**Fig. 3 Identification of CpxR binding sites in the *fabA* and *fabB* promoters. a, b** DNA sequences of *fabA* and *fabB* promoter regions. The FabR box, FadR box and CpxR box sequences were underlined. Two previously reported transcription start sites of *fabA*, S1 and S2, and corresponding −35 and −10 regions were shown in blue and red, respectively. The transcription start site identified in this study and its −35 and −10 regions were shown in green. Numbering is from the start codon of each gene. **c** Western blot analysis of FabA-His$_6$ and FabB-His$_6$ in BW25113 wild-type strain at pH 7.0 and 4.2, and strain with CpxR site substitution at pH 7.0 and 4.2. $n = 2$ biologically independent samples. CB⁻ substituted CpxR box. **d** Relative mRNA level of *fabA* and *fabB* determined by qRT-PCR in BW25113 wild-type strain at pH 7.0 and 4.2, and strain with CpxR site substitution at pH 7.0 and 4.2 ($n = 2$ biologically independent samples with three technical repeats). The *fabA* up primer sequences were shown in purple in (**a**). **e** Acid tolerance of BW25113 wild-type strain and mutants with substitution in *fabA* or *fabB* CpxR box after acidic challenge at pH 4.2 for 1 h ($n = 3$ biologically independent samples). **f** Gel-shift assay. $^{32}$P-labeled DNA fragments containing *fabA* and *fabB* promoter regions were incubated without and with His$_6$-CpxR protein, shown in lanes 1–2. Lane 3 is the same as lane 2 but supplemented with "cold" DNA fragments. **g** DNase I footprinting analysis of *fabA and fabB* promoters with probes for the noncoding strand and increasing amount of His6-CpxR protein. His$_6$-CpxR-DNA mixture were subjected to 5% polyacrylamide electrophoresis, and visualized by autoradiography. The regions protected by CpxR protein were shown in orange in (**a, b**). AG, DNA sequence ladder generated with the same primers using a Maxam and Gilbert A + G reaction. **f** and **g** are representative results from two independent experiments. Error bars, mean ± SEM. Two-tailed Student's *t* tests were performed to determine the statistical significance for two group comparisons. The source data are provided as a Source Data file.

showed that the activating effect of acid stress on expression of *fabA* and *fabB* was completely eliminated by substitution of CpxR site (Fig. 3c, d). Moreover, the acid tolerance of these mutants significantly decreased (Fig. 3e).

The function of CpxR site was further confirmed in vitro through gel-shift assay and DNase I footprinting analysis using purified His$_6$-CpxR protein and 211- and 219-bp DNA fragments corresponding to the promoter regions of *fabA* and *fabB*, respectively. The purified His$_6$-CpxR protein could shift those two DNA fragments (Fig. 3f), and protect the *fabA* promoter at the −287 to −255 region (numbering from the start codon) and the *fabB* promoter at the −119 to −94 region in the noncoding strand containing the putative CpxR binding site (Fig. 3g). All these results demonstrate that the CpxR protein enhances transcription of *fabA* and *fabB* by direct binding to their promoters.

**CpxRA is directly activated by acidic environments.** As a sensor histidine kinase in two-component system, CpxA spans the cell membrane and exposes its sensor domain into the periplasm. When CpxA detects a specific signal, it autophosphorylates and then transports the phosphate group to its cognate regulator CpxR, enabling the regulatory activity of CpxR. Without inducing signal, CpxA acts as a phosphatase to maintain CpxR in an inactive state[35,36].

To monitor the expression and phosphorylation level of CpxA protein, we constructed a strain carrying chromosomal His$_6$-tagged CpxA. As shown in Fig. 4a, acid shock significantly enhanced the amount of CpxA-His$_6$ protein in exponentially growing cells. More importantly, increased level of phosphor-CpxA, the active form of CpxA, was detected at pH 4.2 (Fig. 4a).

To demonstrate direct activation of CpxRA upon exposure to acidic pH, we used a reconstituted proteoliposome system,

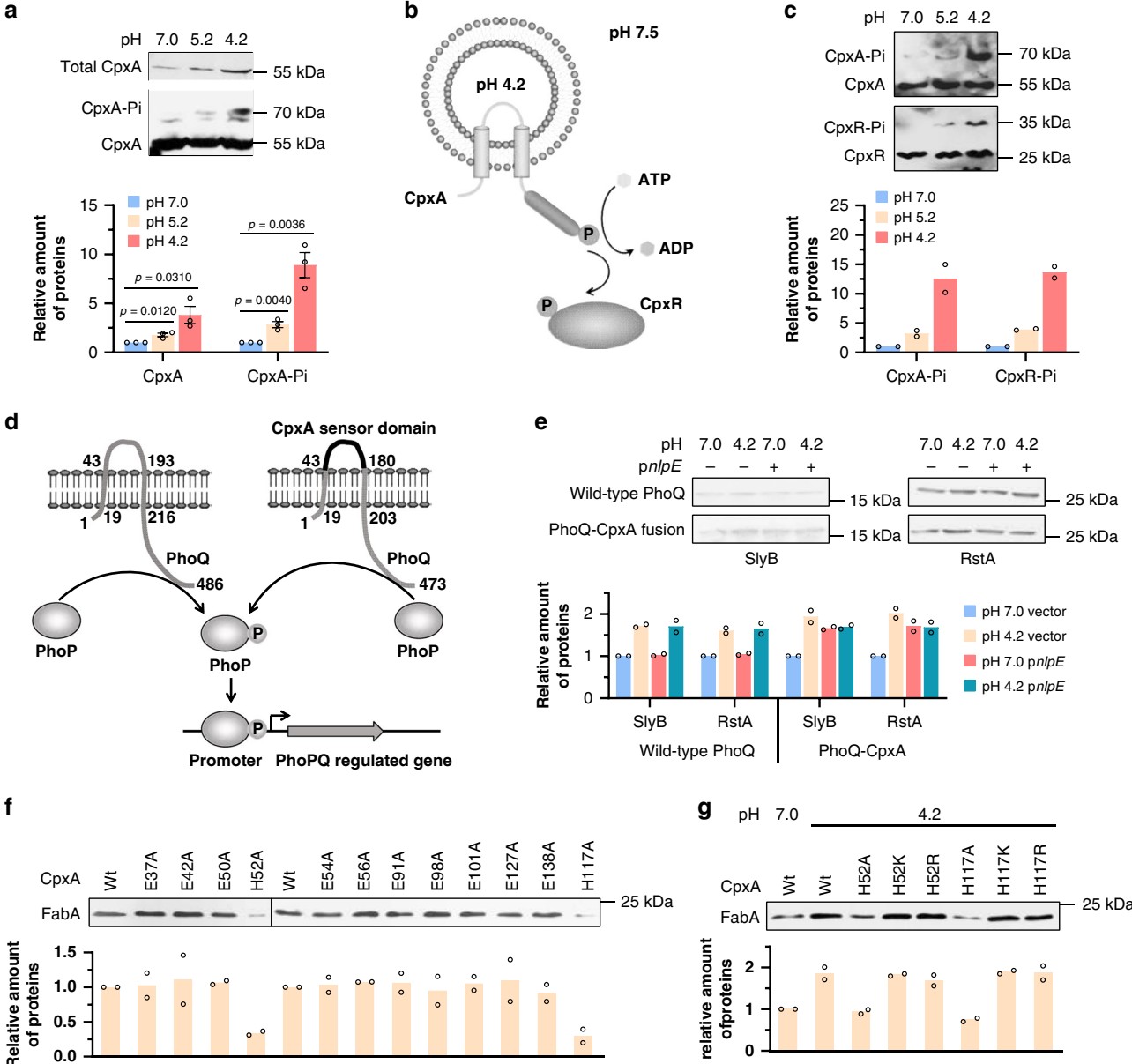

**Fig. 4 The CpxRA system is activated by protonation of CpxA periplasmic histidine residues. a** Western blot of CpxA-His$_6$ protein in BW25113 strain after acidic challenge. For total CpxA protein, the same amount of total cellular proteins extracted from cells grown at different pH was applied to SDS-PAGE. For analysis of phosphorylated CpxA protein, the same amount of CpxA-His$_6$ protein purified from cells grown at different pH was separated using SDS-PAGE containing Mn$^{2+}$ and Phos-tag acrylamide, which retard the mobility of phosphoproteins. $n = 3$ biologically independent samples. **b** Schematic diagram of the proteoliposome system. **c** In vitro analysis of CpxA and CpxR phosphorylation induced by acidic pH, carried out using reconstituted proteoliposomes, which were preloaded with buffers at different pH and purified CpxA-His$_6$ protein. To analyze phosphotransfer, purified His$_6$-CpxR was incubated with CpxA-His$_6$-containing proteoliposome in phosphorylation buffer. $n = 2$ biologically independent samples. **d** Design and construction of a PhoQ-CpxA fusion protein. PhoPQ is a known two-component system which can sense the acidic pH, and activate transcription of *rstA* and *slyB* genes. We replaced the sensor domain of PhoQ (amino acids 43–193) with the periplasmic domain of CpxA (amino acids 28–164). **e** Western blot of RstA and SlyB proteins in BW25113 with wild-type PhoQ or PhoQ-CpxA fusion at pH 7.0, at pH 4.2, with NlpE overexpression, and at pH 4.2 with NlpE overexpression. $n = 2$ biologically independent samples. **f** Western blot of FabA-His$_6$ protein in BW25113 strains carrying wild-type CpxA or CpxA mutants in which the periplasmic histidine/glutamate was replaced with alanine at pH 4.2. $n = 2$ biologically independent samples. **g** Western blot of FabA-His$_6$ protein in BW25113 strains carrying wild-type CpxA or CpxA mutants in which the histidine at positions 52 and 117 was replaced by alanine, lysine, and arginine, respectively. $n = 2$ biologically independent samples. Error bars, mean ± SEM. Two-tailed Student's $t$ tests were performed to determine the statistical significance for two group comparisons. The source data are provided as a Source Data file.

where a different internal and external pH can be stably maintained (Fig. 4b). The CpxA-His$_6$ protein was purified from membranes and reconstituted into vesicles mainly in the inside-out orientation using the detergent-mediated method as described[37]. As shown in Fig. 4c, lowering the pH inside

vesicles from 7.0 to 4.2 greatly enhanced the amount of phospho-CpxA. The phosphoryl group was transferred to regulator CpxR when the pH inside vesicles was 4.2, while phosphorylated CpxR could be hardly detected with neutral lumen pH (Fig. 4c) or without the sensor kinase CpxA. These

results suggested that CpxA is capable of activating CpxR upon direct exposure to acidic environments.

To further verify the sensitivity of CpxA periplasmic domain to acidification, a chromosomal PhoQ-CpxA fusion strain was constructed, in which the sensor domain of PhoQ (amino acids 43–193) was replaced by the periplasmic domain of CpxA (amino acids 28–164) (Fig. 4d). PhoPQ is a known two-component system that can sense the acidic pH, $Mg^{2+}$ depletion and antimicrobial peptides, and activate transcription of genes including *rstA* and *slyB*[38,39]. Western blot results showed that both low pH and overexpression of NlpE promoted expression of RstA and SlyB in strain carrying the PhoQ-CpxA fusion protein, while only acidic environments increased the protein level of RstA and SlyB in wild-type strain (Fig. 4e). On the other hand, when the CpxA periplasmic domain was replaced with that from another kinase AtoS which senses the presence of acetoacetate[40], neither could the protein level of FabA-His$_6$ and FabB-His$_6$ be enhanced by acidic conditions in vivo, nor could the CpxR protein be phosphorylated at pH 4.2 in reconstituted proteoliposome (Supplementary Fig. 3), indicating that CpxA periplasmic domain is sensitive to acidification. All these results collectively demonstrated that CpxRA system is activated by direct exposure of CpxA periplasmic domain to acidic environments.

**CpxA H52 and H117 are required for sensing of acidic pH**. As reported, the histidine and glutamic acid residues were regarded as sensors detecting mild acidic pH because they can change their protonation state upon variation in the surrounding pH[3,41–43]. So, we hypothesized that the histidine and glutamate residues in CpxA sensor domain might be required for pH sensing. To test this hypothesis, we constructed a series of plasmids encoding CpxA mutants in which the periplasmic histidine/glutamate residue was replaced with alanine, respectively. Strains expressing the mutant CpxA proteins could express *fabA* in response to NlpE overexpression normally (Supplementary Fig. 4), indicating that mutations in residues of the periplasmic domain of CpxA protein do not impair the kinase activity of CpxA cytoplasmic domain. As shown in Fig. 4f, only two mutants (H52A and H117A) resulted in the loss of CpxA capability to upregulate *fabA* expression at pH 4.2, while the other mutations in glutamate residues did not change the expression level of *fabA*. To confirm that protonation of H52 and H117 is responsible for pH-mediated gene regulation, we also mutated these two residues to other basic amino acids, lysine and arginine, respectively. It was discovered that the substitution of H52 and H117 by lysine and arginine had no effect on the response to acidic pH (Fig. 4g). Moreover, the strain carrying CpxA variant with a basic amino acid residue at positions 52 or 117 presented a higher level of FabA protein even at pH 7.0, when compared with strain with CpxA H52A or H117A mutant (Supplementary Fig. 5), probably due to the protonation of basic amino acid residue at pH 7.0. These results demonstrate that protonation of residues at positions 52 and 117 is required to maintain the response of CpxA to acidic pH, but the presence of a histidine at those positions is not a specific requirement for activation.

**Physiological effects of changes in lipid composition**. To learn how changed content in membrane lipid affect *E. coli* cell functions and physiology, several assays were carried out using cells carrying empty vector as control, p*fabA*, or p*fabB*, after acidic challenge at pH 4.2. The membrane fluidity was determined by analyzing the fluorescence anisotropy of 1,6-diphenyl-1,3,5-hexatriene (DPH), which is negatively correlated to fluidity[44]. As shown in Fig. 5a, the strains with overexpression of *fabA* or *fabB* presented much less cell membrane fluidity than strain carrying empty vector. The membrane permeability was measured by detecting the leakage of $OD_{260}$ materials (predominantly nucleotides), and there was no significant difference observed despite the varied UFA contents in membrane (Supplementary Fig. 6). $F_0F_1$-ATPase spans the cell membrane and transports periplasmic protons to cytoplasm with production of ATP, and lipids are required for its optimal functioning[45]. The activity of $F_0F_1$-ATPase was measured, and results suggested that the enhancement of UFAs content repressed its activity (Fig. 5b). Then the intracellular pH was monitored using ratiometric pH-sensitive GFP pHluorin2[46]. When growing under pH 5.0–9.0, *E. coli* can maintain the intracellular pH between pH 7.4 and 7.9[47,48]. Upon exposure to pH 4.2, the internal pH of control strain decreased to $6.82 \pm 0.06$, and the homeostasis of pH in *E. coli* cells with overexpression of *fabA* or *fabB* was good (Fig. 5c), probably due to the lowered membrane fluidity and $F_0F_1$-ATPase activity.

As *E. coli* has to combat a moderate acidic environment (pH 4.0–6.0) and proliferate in host's small intestine, we carried out mouse gastrointestinal passage experiment to test the in vivo effect of the UFAs-CpxRA system. BALB/c mice were administered BW25113 wild-type strain or mutant with substituted *fabA* CpxR binding site, and three independent trials with six mice in each trial were performed for each strain. After 24 h, fecal samples were collected to test the presence of each strain. Out of 18 inoculated mice, the wild-type BW25113 strain was recovered from a total of 8 mice, while the *fabA* CpxR box mutant was detected in only 2 fecal samples (Fig. 5d). The results between the parental and mutant strains were statistically different ($p < 0.05$), demonstrating that this ATR system significantly promotes *E. coli* survival in mouse intestinal lumens.

In bio-production of organic acids, product accumulation acidifies the fermentation broth and inhibits the growth of producing strain. Previously, we had constructed a 3-hydroxypropionate (3HP)-producing *E. coli* recombinant strain. In shaking flask cultivation, the pH of culture decreased to pH 5.2–5.5 along with the production of 3HP, repressing the further production and cell growth (Fig. 5e). So, the pH of fermentation broth had to be adjusted to 7.0 periodically. Then *fabA* gene was overexpressed to enhance *E. coli* tolerance to acidic environments, leading to similar 3HP production and cell growth with and without pH adjustment (Fig. 5e). This result demonstrates the great prospect of UFAs-CpxRA-dependent ATR system in organic acids bio-production.

**The UFAs-CpxRA system functions in diverse bacteria species**. *Salmonella* Typhimurium LT2 and *Shigella flexneri* 2a str. 2457T are both common enteric pathogens, and their FabA proteins share more than 98% identity with *E. coli* FabA (Supplementary Fig. 7). To test whether UFAs are required for growth under acidic pH in these bacteria, the empty vector and recombinant plasmid carrying *E. coli fabA* gene were transformed into *Salmonella* Typhimurium LT2 and *S. flexneri* 2a str. 2457T. In acidic challenge tested at pH 5.0, the transformation of empty vector did not affect the acid tolerance of those two strains, but the overexpression of *fabA* gene elevated those CFU ratios by 2.32- and 1.99-fold, respectively (Fig. 6a). Moreover, mRNA level of *fabA* and *fabB* was upregulated by acid challenge in *Salmonella*, *Klebsiella* and *Pseudomonas* strains (Fig. 6b), and the consensus sequence of CpxR site was also found upstream of the *fabA* and *fabB* genes in additional bacteria species (Table 1). Additionally, *Cronobacter sakazakii* clone carrying transposon insertion in *cpxR* gene was identified as acid-sensitive mutant[49]. All these facts suggest this ATR system functioning in exponential phase is highly conserved across bacteria species.

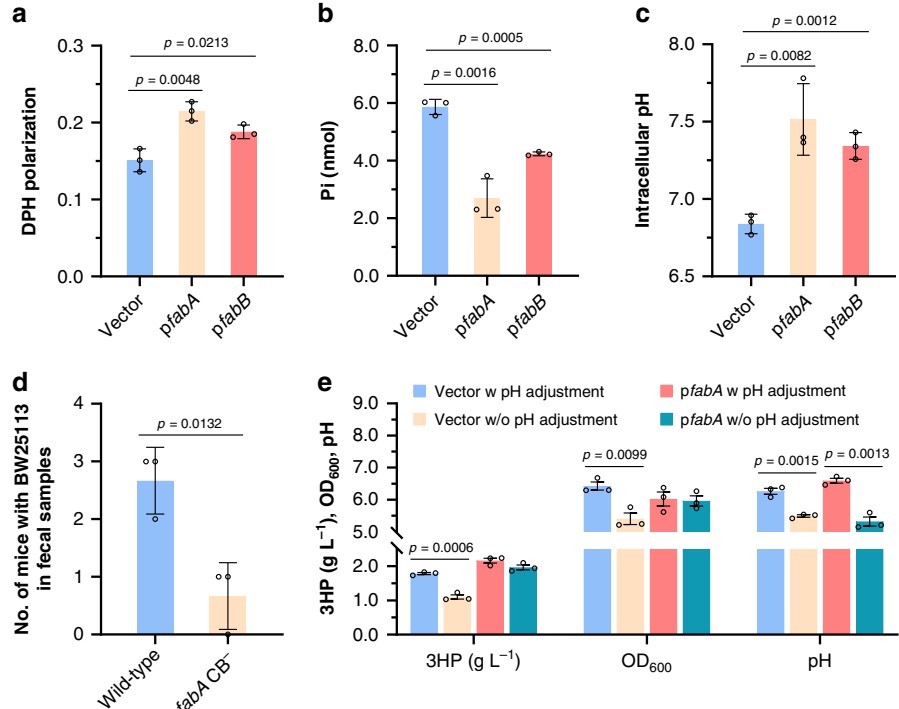

**Fig. 5 The UFAs content affects membrane properties, _E. coli_ survival in host's intestine and production of organic acid.** The membrane fluidity (**a**), $F_0F_1$-ATPase activity (**b**), and intracellular pH (**c**) of BW25113 strains carrying empty vector pTrcHis2B, p_fabA_ or p_fabB_, after acidic challenge at pH 4.2 for 1 h. The membrane fluidity is determined by the anisotropy of fluorescent probe 1,6-diphenyl-1,3,5-hexatriene (DPH). $F_0F_1$-ATPase activity is assayed in terms of the release of inorganic phosphate using permeabilized cells in ATP-containing buffer. The intracellular pH was monitored using ratiometric pH-sensitive GFP pHluorin2. **d** Recovery of _E. coli_ BW25113 wild-type strain and mutant with substitution in _fabA_ CpxR site from feces following oral administration to BALB/c mice. Three independent trials with six mice in each trail were performed for each strain. **e** 3-Hydroxypropionate production, $OD_{600}$, and final pH of fermentation broth of recombinant strains with and without _fabA_ overexpression. For experiments with pH adjustment, the medium was adjusted to pH 7.0 every 12 h. Data are presented as mean ± SEM of three independent experiments. Two-tailed Student's _t_ tests were performed to determine the statistical significance for two group comparisons. The source data are provided as a Source Data file.

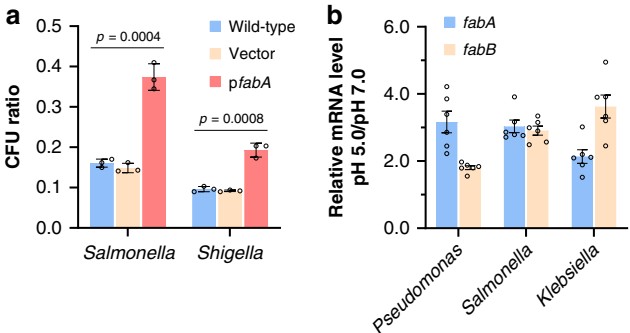

**Fig. 6 The exponential phase ATR system based on UFAs and CpxRA is conserved in bacteria. a** Acid tolerance of _Salmonella_ Typhimurium LT2 and _Shigella flexneri_ 2a str. 2457T wild-type strain and strains carrying empty vector or p_fabA_ after acidic challenge at pH 5.0 for 1 h ($n = 3$ biologically independent samples). **b** Relative mRNA level of _fabA_ and _fabB_ in _Salmonella_ Typhimurium LT2, _Klebsiella pneumoniae_ ATCC25955 and _Pseudomonas aeruginosa_ PAO1 grown at pH 7.0 and pH 5.0 ($n = 2$ biologically independent samples with three technical repeats). Error bars, mean ± SEM. Two-tailed Student's _t_ tests were performed to determine the statistical significance for two group comparisons. The source data are provided as a Source Data file.

## Discussion

According to the results shown above, we present a previously uncharacterized ATR system as well as its regulatory pathway in exponentially growing _E. coli_. We demonstrated that the

two-component system CpxRA can directly sense the acidic environments, and activate transcription of UFAs synthetic genes, resulting in increased UFAs contents in cell membrane lipid and normal growth of _E. coli_ at pH 4.2. Our findings greatly enriched our understanding of the networks contributing to bacterial acid resistance, and the mechanisms for stress response governed by the CpxRA system.

As reported, the acid limit for _E. coli_ growth is pH 4.0–4.5, and all known AR and ATR systems only prolong survival of _E. coli_ cells under acidic conditions, but cannot support growth at pH 4.0–4.5[18,21,22]. In our study, overexpression of _fabA_ and _fabB_ genes for UFAs biosynthesis, or activation of the two-component regulatory system CpxRA both restore the growth capability of _E. coli_ at pH 4.2. Compared with those previously known AR systems, this growth-conferring ATR system is expected to have more important physiological significance. Firstly, bacteria can grow normally under acidic pH with activation of the CpxRA- and UFAs-dependent system, achieving higher biomass, which may be required for successful pathogenesis and efficient bio-production. Secondly, de novo mutations during DNA duplication play a critical role in bacterial stress resistance development[50], and the rapid proliferation at low pH provides an opportunity for the evolution of novel AR and ATR systems. Thirdly, functioning of previously known AR systems is subject to more external limitations, such as AR1 is repressed by glucose and AR2−AR5 are dependent on exogenous amino acids, whereas the UFAs-CpxRA system can be activated by acidic pH alone because CpxA is capable of phosphorylating CpxR upon exposure to pH 4.2 in a reconstituted proteoliposome system.

**Table 1 Putative CpxR box of *fabA* and *fabB* genes from Gram⁻ bacteria (The genome sequences of various bacteria used in this table are under the accession numbers CP009273, NC_003197, NC_002516, NC_011283, NC_013716, CP001235, NC_004741, CP001589, and NC_009778, respectively).**

| Bacteria species | *fabA* gene | | *fabB* gene | |
| --- | --- | --- | --- | --- |
| | Sequence[a] | Position[b] | Sequence[a] | Position[b] |
| *E. coli* BW25113 | GTAGAagaagGCAAA (8) | −272 to −258 | GTAAGgctgcGCAAA (8) | −112 to −98 |
| *Salmonella typhimurium* LT2 | GTAAAcagggcGTGAT (8) | −343 to −328 | GCAAAtttccGTACT (8) | −103 to −89 |
| *Pseudomonas aeruginosa* PAO1 | GGAATtgcaacGCATA (6) | −279 to −264 | GTCGAtggccGCGAA (6) | −83 to −69 |
| *Klebsiella pneumoniae* 342 | GTAAAcagggcGTCAT (8) | −341 to −326 | GCAAAtttatACACA (6) | −102 to −88 |
| *Citrobacter rodentium* ICC168 | GTAAGcagggGTAAT (8) | −343 to −328 | GCAAAtttcgGCAGA (7) | −102 to −88 |
| *Vibrio cholerae* O395 | ATAAAtcagtaGTAGA (8) | −282 to −267 | GAAATcaggctGAATA (6) | −110 to −95 |
| *Shigella flexneri* 2a str. 2457 T | GTAGAagaagGCAAA (8) | −272 to −258 | GTAAGgctgcGCAAA (8) | −112 to −98 |
| *Yersinia pestis* D182038 | GTACAgaaagGTCAA (8) | −269 to −255 | GTAAAattagtGGAAG (8) | −25 to −10 |
| *Cronobacter sakazakii* ATCC BAA-894 | GCAAAcagggcGTGAT (8) | −340 to −325 | GCAAAtttcttGCCAA (7) | −101 to −86 |

[a]Number of bases same with the CpxR site consensus was presented in the parenthesis.
[b]Numbering is from the start codon of *fabA* or *fabB* gene.

We proved that the transcription of *fabA* stimulated by CpxRA is required for multiplication of *E. coli* in mouse intestine, indicating that UFAs-CpxRA system is also related to pathogenesis of *E. coli* pathovars, as well as those previously known AR systems. While various pathotypes of *E. coli* colonize and infect different organs, they all have to combat acidic environments during invading the host's digestive tract. With pH values as low as 1.5–2.5, the stomach has been recognized as a natural antibiotic barrier[1]. Benefited from AR1−AR5 systems, *E. coli* can survive in the gastric acid for hours[22,51]. With their passage into the small intestine, *E. coli* cells will encounter a less acidic environment (pH 4.0–6.0) with the presence of organic acids produced by the normal intestinal flora[2]. As pathogenic *E. coli* strains must reproduce rapidly to cause disease ultimately[52], the UFAs-CpxRA system is likely to play a key role. In summary, the successful enteric pathogen must possess two abilities, survival in extreme acidic condition and quick growth in moderate acidic environment. Consequently, this ATR system may be a new target for the development of antimicrobials.

UFAs-CpxRA-dependent ATR system also has potential application in bio-production of organic acids, which are valuable platform chemicals and have been successfully produced by recombinant *E. coli* strains[53,54]. However organic acids cause acidification of fermentation broth and inhibit *E. coli* growth at concentrations far below what is required for economical production. Now large quantity of base titrant are required to raise pH of the media in organic acids production process, and large amounts of acid must be consumed to recover the organic acids in the protonated form after production. If we could construct acid-tolerant strains growing at a pH less than the pKa of the produced acid, the additional consumption of acid and base titrants will be circumvented and the overall production cost will be lowered remarkably. As the UFAs-CpxRA system functions in exponential phase, is not repressed by glucose (the carbon source in most fermentation), and does not need exogenous amino acids, it is believed that this ATR system could be effectively applied in the field of organic acids bio-production.

Our data suggested that UFAs played an important role in protection of exponential phase *E. coli* from acid shock. The increase of UFAs content in membrane lipid not only affected the fluidity of lipid bilayer but also changed the activity of the $F_0F_1$-ATPase, conducing to reduced membrane proton permeability and improved internal pH homeostasis. This phenomenon is consistent with that *S. mutants* cells grown at pH 5.0 had higher UFAs composition and lower proton permeability than those grown at pH 7.0[27,55]. Moreover, changes in fatty acid composition probably also affect the PTS system and enzyme secretion[56,57]. Overall, changing membrane fatty acid composition may improve the bacterial ability to adapt to acidic environment and be an important factor in bacterial acid response. In this study, the cyclopropane fatty acid (CFA) could not be detected in exponential phase *E. coli* cells, although it was regarded as a major factor in acid resistance of stationary phase *E. coli*[24,58]. However, UFA can be converted into CFA by CFA synthase[59], and increased UFAs content will potentially enable the synthesis of CFA in stationary phase.

We demonstrate that the *E. coli* kinase CpxA is a direct sensor for acidic pH. Our data provide strong evidence that a decrease in pH protonates the histidine residues at positions 52 and 117 in CpxA periplasmic domain, leads to events catalyzed by its cytoplasmic domain, including phosphorylation of CpxA, transfer of phosphoryl group to regulator CpxR, as well as activation of CpxR-dependent gene transcription. As titratable by pH, histidine has been regarded as sensor detecting mild acidic pH, and also plays an essential role in the activation of sensor kinase PhoQ by acidic pH[60]. But the molecular details of how histidine senses the low pH signal in PhoQ and CpxA are different. Protonation of residues at positions 52 and 117 is effective to activate CpxA at acidic pH, whereas the imidazole ring of histidine is important in maintaining the response of PhoQ to acidity[60].

Our results evidenced that CpxRA is a key system in the acid stress response of exponentially growing *E. coli* (Fig. 7), consistent with a previous proteomic analysis highlighting the importance of CpxRA in acid stress[61]. Upon exposure to acidic environments, CpxRA system stimulates the transcription of UFAs synthetic genes, resulting in improved intracellular pH homeostasis. Additionally, CpxRA upregulates some genes involved in cell wall modification, including peptidoglycan (PG) cross-linking proteins YcfS, YcbB and DacC; PG cleaving proteins AmiA, AmiC and Slt[61–63]. The induction of those genes led to an increase of cross-linking between PG and outer membrane proteins, and an increase of cell wall stability, which may help protecting *E. coli* cells from acidic challenge. That proteomic study also indicated the repression of AR2 system by CpxRA[61]. Because AR2 system is responsible for survival below pH 3.0 and log phase cells with overexpression of AR2 genes are not more acid resistant[4], the CpxRA-mediated repression of AR2 in exponentially growing cells above pH 3.0 will guarantee that AR2 system is not induced in an inappropriate situation to avoid the metabolic burden. Furthermore, the sensor kinase CpxA was proved to cross-talk with noncognate response regulator OmpR[64], which itself is involved in the acid stress response of *E. coli*[65,66]. All of these data indicate that *E. coli* has

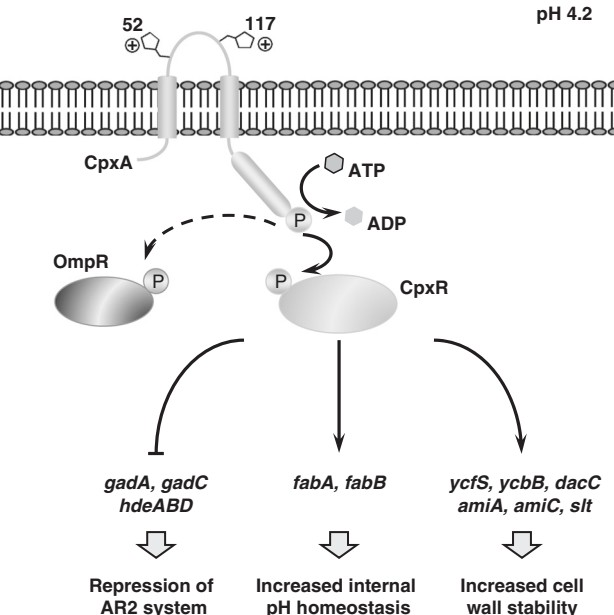

**Fig. 7 Model illustrating the CpxRA-dependent gene regulation involved in exponential phase ATR system.** In exponentially growing *E. coli*, CpxRA system is activated by moderate acidic pH through protonation of periplasmic histidine residues, and upregulates the transcription of UFAs biosynthesis genes *fabA* and *fabB*, and cell wall modification genes including *ycfS*, *ycbB*, *dacC*, *slt*, *amiA* and *amiC*, and represses the expression of AR2 system genes. The sensor kinase CpxA also interacts with noncognate response regulator OmpR, which itself plays an important role in *E. coli* acid resistance.

evolved multiple acid response mechanisms as well as precise regulatory circuits, to target specific stress conditions.

In many Gram-negative pathogens, the CpxRA system plays an important role in the regulation of virulence factors, including pilus, secreted virulence effectors and type III secretion system[36]. For CpxRA to upregulate expression of these virulence genes, there must be an activating signal for CpxA under in vivo conditions. The identity of this signal still remains a topic of much debate. According to the results in this study, we propose that moderate acidic pH (such as the intestine and macrophage) is the activating signal of CpxA in vivo. As the previous study showed that CpxRA also contributed to bacterial resistance to antimicrobial peptides, a component of the host antimicrobial response[62], acidic pH and antimicrobial peptide may have synergistic effect on CpxRA activation. Further study is ongoing to address this hypothesis.

## Methods

**Bacterial strains and growth conditions**. All strains and plasmids used in this study are listed in Supplementary Data 1, and all primers used are listed in Supplementary Data 2. Phage P1 was used for generalized transductions in *E. coli*. Bacteria were grown at 37 °C in Luria-Bertani broth (Oxoid) or in E minimal medium (0.8 mM MgSO$_4$, 10 mM citric acid, 57.5 mM K$_2$HPO$_4$, 16.7 mM NaNH$_3$HPO$_4$, 0.5% glucose). When necessary, antibiotics were added at final concentrations of 100 μg mL$^{-1}$ for ampicillin, 20 μg mL$^{-1}$ for chloramphenicol or 50 μg mL$^{-1}$ for kanamycin. *E. coli* DH5α was used as host for the preparation of plasmid DNA, and *E. coli* χ7213 was used for preparation of suicide vectors. Diaminopimelic acid (50 μg mL$^{-1}$) was used for the growth of χ7213 strain. LB agar containing 10% sucrose was used for *sacB* gene-based counter selection in allelic exchange experiments. The software ImageJ (version 1.52a) was used to analyze western blot results, and the online version of Clustal Omega (https://www.ebi.ac.uk/Tools/msa/clustalo) was used for sequence alignment.

Plasmids were constructed by digesting PCR fragments containing target gene and cloning into corresponding vectors as normal. Derivatives of pTrc-*cpxA* with nucleotide substitutions were constructed using Q5 Site-Directed Mutagenesis Kit (New England Biolabs) according to the manufacturer's specifications. All plasmids

were confirmed by DNA sequencing. Strains harboring chromosomal epitope-tagged proteins were generated using λ Red recombinase system[67,68]. DNA fragments encoding PhoQ-CpxA fusion or carrying substituted CpxR site were generated by joint PCR using primers shown in Supplementary Data 2, and cloned into the suicide vector pRE112[69]. The resulting plasmids were used to mediate the allelic exchange to generate strains with chromosomal PhoQ-CpxA fusion and CpxR site mutation.

In acidic challenge experiments, the strains were grown in E medium (pH 7.0) to an OD$_{600}$ of 0.6. For strains with plasmid, IPTG was added to final concentration of 0.5 mM at an OD$_{600}$ of 0.4, and the strain was further grown to an OD$_{600}$ of 0.6. Then, the cells were harvested and washed twice with fresh E medium (pH 7.0), and inoculated into E medium with various pH as indicated, and strains were grown for another 30–60 min before the cells were collected to determine the CFU, membrane lipid composition, mRNA and protein levels.

**Determination of fatty acid compositions**. Phospholipids were extracted as described by Wang and Cronan[23]. Briefly, *E. coli* cells were harvested and resuspended in 1 mL sterile water, and 5 mL of chloroform-methanol (2:1 vol/vol) was added and vortexed for 3 min then stewing overnight. The solution was centrifuged and the upper phase was removed, an equal volume of 2 M KCl was added, followed by mixing and centrifugation. The top KCl phase was removed, an equal volume of water was added, followed by mixing and centrifugation. The resulting bottom organic phase was dried under a stream of nitrogen. Two milliliters of methyesterification reagent (BF$_3$:CH$_3$OH = 1:4) was added and incubated at 60 °C for 30 min. The resulting solution was extracted with n-hexane for three times and then detected with Agilent GC-MS (7890A–5975C; Column: Agilent-HP-INNO-Wax). The fatty acids were identified by comparing the retention times and mass fragmentation patterns with authentic standards. Content of each fatty acid is given as the relative peak area [(peak area of one fatty acid/total peak area) × 100%].

**Immunoblot analysis of His$_6$-tagged proteins**. The *E. coli* cells were disrupted by sonication, the cell lysates were centrifuged, and the supernatants were used for western blot. Protein concentration was determined using BCA protein assay kit (Pierce), and the same amount of protein sample was separated in 12% SDS-PAGE, transferred to nitrocellulose membranes (Bio-Rad), and incubated with monoclonal HRP-conjugated anti-6×His antibodies (1:10,000 diluted) (Abcam). Protein signals were detected using Immobilon Western HRP substrate (Millipore) and X-ray film. To separate the phosphorylated proteins, 20–50 μM Phos-tag Acrylamide (WAKO) and 0.1 mM Mn$^{2+}$ were added into the SDS-PAGE. Quantification was conducted using ImageJ software (NIH).

**Quantitative RT-PCR and RACE**. Total RNA was isolated from bacterial culture using EASYSpin Plus bacterial RNA quick extract kit (Aidlab Biotechnologies, China) according to the manufacturer's instructions. RNA concentration was determined by spectrophotometry at 260 nm. Removal of genomic DNA and synthesis of cDNA were carried out using PrimeScript RT reagent Kit with gDNA Eraser (Takara). qRT-PCR was conducted using TB Green Premix Ex Taq (Takara) with the QuantStudio 1 system (Applied Biosystems). Constitutively transcribed gene *rpoD* was used as a reference control to normalize total RNA quantity of different samples. The relative difference of mRNA level was calculated using the ΔΔCt method[70]. Two independent biological samples with three technical repeats for each sample were performed for each qRT-PCR analysis.

The RACE experiment was performed using SMARTer RACE cDNA Amplification Kit (Clontech), according to the manufacturer's instructions. The primers 1128 + 1129 and 1130 + 1131 were used to determine the transcription start sites of *fabA* and *fabB* under the NlpE overexpression conditions, respectively.

**Purification of His$_6$-CpxR and CpxA-His$_6$**. Purification of His$_6$-CpxR and CpxA-His$_6$ was conducted according to Fleischer et al.[37]. Briefly, *E. coli* strain BW25113 with pTrc-*cpxR* or pTrc-*cpxA* was grown at 37 °C with aeration in LB medium. Gene expression was induced with 0.5 mM IPTG for 3–4 h. Membrane fractions and cytosolic fraction were separated by ultracentrifugation[37]. His$_6$-CpxR was solubilized in cytosolic fraction and purified by Ni-affinity chromatography. Membrane proteins were solubilized with 1% dodecyl-β-D-maltoside (DM), and then CpxA-His$_6$ was also purified by Ni-affinity chromatography.

**Preparation of proteoliposomes**. Proteoliposome was reconstructed as previously described with small modification[37]. Briefly, *E. coli* phospholipids (Avanti) were dried under a stream of nitrogen, and slowly dissolved in sodium citrate-hydrochloric acid buffer (pH 7.0, 5.0, 4.2) with 10% glycerol (vol/vol) and 0.47% Triton X-100 (vol/vol), respectively. Purified CpxA-His$_6$ was added to the mixture and stirred at room temperature for 20 min. Bio-Beads SM-2 (Bio-Rad) were added in a bead/detergent ratio of 10:1(w/w), and the mixture was gently stirred at 4 °C overnight. After 16 h, fresh Bio-Beads were added, and the mixture was stirred for another 6 h. The proteoliposomes were collected by ultracentrifugation. To test autophosphorylation, proteoliposomes were incubated with 300 μmol ATP in phosphorylation buffer (50 mM Tris-HCl, pH 7.5, 10% glycerol (vol/vol), 2 mM dithiothreitol (DTT), 50 mM KCl, 5 mM MgCl$_2$) at room temperature for 30 min. 5× SDS sample buffer was loaded to termination reaction. To analyze

phosphortransfer, purified His$_6$-CpxR was added to this mixture and incubated at room temperature for 20 min. Then samples were ultracentrifugated and the upper phase was collected. 5× SDS sample buffer was loaded to stop the reaction. To detect the phosphorylation level of CpxA and CpxR, all the samples were subjected to 8% SDS-PAGE with 20–50 μM Phos-tag Acrylamide (WAKO) and 0.1 mM Mn$^{2+}$, which can retard the mobility of phosphoproteins to show the phosphorylated and nonphosphorylated forms in two separated bands.

**Electrophoretic mobility shift assay (EMSA)**. Primers 683 and 685 were labeled using T4 polynucleotide kinase (New England Biolabs) and γ-$^{32}$P ATP (Perkinelmer Life Sciences). The promoter regions of *fabA* and *fabB* genes were amplified with primers 682 + 683 and 684 + 685, respectively. Ten nmol of $^{32}$P-labeled DNA was incubated at room temperature for 30 min with 0 or 50 pmol of His$_6$ -CpxR protein in 20 μL of an EMSA buffer consisting of 10 mM Tris-HCl, pH 7.5, 1 mM ethylene diamine tetraacetic acid, 5 mM DTT, 10 mM NaCl, 1 mM MgCl$_2$, and 5% glycerol. The mixture was subjected directly to 4% TAE-PAGE. Signals were detected by autoradiography.

**DNase I footprinting assay**. DNase I footprinting assays were carried out using the *fabA* promoter region amplified from BW25113 chromosome with primers 682 and $^{32}$P-683 and using the *fabB* promoter region amplified with primers 684 and $^{32}$P-685 for the noncoding strand. Approximately 25 pmol of $^{32}$P-labeled DNA and 0, 25, 50, or 100 pmol of His$_6$-CpxR protein were mixed in a 100-μL reaction containing 20 mM 2-[4-(2-hydroxyethyl)piperazin-1-yl]ethanesulfonic acid pH 8.0, 10 mM KCl, 1 mM DTT, and 0.1 mg mL$^{-1}$ bovine serum albumin. The reaction mixture was incubated at room temperature for 20 min. Then 1 μL of 100 mM CaCl$_2$, 1 μL of 100 mM MgCl$_2$, and 0.005 units of DNase I (Fermentas) were added, and the mixture was incubated at room temperature for 2 min. The DNase I digestion was stopped by phenol treatment, and the DNA was precipitated. Samples were analyzed by 6% polyacrylamide electrophoresis by comparison with a DNA sequence ladder generated with the same primers using a Maxam and Gilbert A + G reaction.

**Membrane property assays**. The membrane fluidity was measured by applying DPH as a fluorescence probe[44]. Briefly, the washed cells were incubated in DPH (Sigma) at a final concentration of 2 μM, then shaken in the dark at 30 °C for 40 min. After incubation, the unincorporated DPH was removed by washing with phosphate-buffered saline (PBS) twice. The cells were resuspended in PBS to get a final density of OD$_{600}$ = 0.2 for the measurement of fluorescence anisotropy on a FluoroMax-4 spectrofluorometer (HORIBA Jobin Yvon). The wavelengths were 342 and 432 nm and slit widths were 5 and 10 nm for excitation and emission light, respectively. The fluorescence anisotropy, which was negatively correlated with membrane fluidity, was calculated[44]. For the permeability assay, the washed cells were resuspended in PBS for 30 min. The release of nucleotides was then measured at an optical density of 260 nm.

**Intracellular pH measurement**. The gene encoding ratiometric pH-sensitive green florescent protein pHluorin2 was synthesized and cloned into vector pBAD-18kan. Excitation assays were performed at wavelengths 395 and 475 nm with the emission at 510 nm using FluoroMax-4 Spectrofluorometer (HORIBA Jobin Yvon), and the ratio of fluorescence at 395 to 475 nm was used to calculate the intracellular pH according to a standard curve[46].

**F$_0$F$_1$-ATPase activity assay**. The cells were washed once and resuspended in 1.8 mL membrane buffer (75 mM Tris pH 7.0, 10 mM MgSO$_4$). Toluene was added to a final concentration of 10% (vol/vol), and the suspension was vortexed for 30 s and subjected to two rounds of freeze-thawing. Cells were collected, resuspended in membrane buffer. ATPase activity was assayed in terms of the release of inorganic phosphate in 50 mM Tris-maleate buffer pH 7.0, containing 10 mM MgSO$_4$ and 50 μM ATP. Phosphate was assayed by the malachite green method, using Malachite Green Phosphate Assay Kit (Cayman Chemical).

**Mouse survival passage experiments**. Female BALB/c mice were housed individually in pathogen-free facility and given food and water ad libitum until 6-week-old. BW25113 strain and mutant with substitution in *fabA* CpxR site were grown in LB at 37 °C to log phase, and then 10$^4$ CFUs of bacteria were orally administrated to mice. To allow for inoculum clearance through the stomach, the mice were not provided feed for 4 h after oral administration. Fecal samples were collected 24 h after inoculation and suspended in PBS (0.5 g feces/4.5 mL PBS) and subsequently diluted. The diluted samples were then plated on LB agar containing nalidixic acid (20 μg mL$^{-1}$) to test for the presence or absence of corresponding strains. Three independent trials were performed for each strain, and in each trial six mice were inoculated. This experiment was performed following the Guide for the Care and Use of Laboratory Animals (National Institutes of Health, 1985).

**3-Hydroxypropionate production**. The strains were grown overnight in LB broth and 1:100 diluted into 250 mL Erlenmeyer flasks with 50 mL of minimal medium containing 14 g L$^{-1}$ K$_2$HPO$_4$•3H$_2$O, 5.2 g L$^{-1}$ KH$_2$PO$_4$, 1 g L$^{-1}$ NaCl, 1 g L$^{-1}$ NH$_4$Cl, 0.5 g L$^{-1}$ MgSO$_4$, 0.2 g L$^{-1}$ yeast extract, and 20 g L$^{-1}$ glucose. All shake flask experiments were carried out in triplicates. After incubation at 37 °C, 0.05 mM IPTG was added for induction at OD$_{600}$ 0.8, and 3 h later, biotin (40 mg L$^{-1}$) and NaHCO$_3$ (20 mM) were added. The antibiotics were supplied periodically after induction of IPTG every 12 h until 48 h. 10 g L$^{-1}$ glucose was added once again after 24 h induction. 3HP concentration in medium was determined using Agilent 1200 Infinity HPLC system with an Aminex HPX-87H column (300 × 7.8 mm, Bio-Rad)[54].

**Ethics declarations**. The animal experiments were performed according to the standards set forth in the Guide for the Care and Use of Laboratory Animals (National Institutes of Health, 1985). Experimental protocols were approved by the Institutional Animal Care Committee at Qingdao Institute of Bioenergy and Bioprocess Technology.

**Reporting summary**. Further information on research design is available in the Nature Research Reporting Summary linked to this article.

## Data availability

The source data underlying Figs. 1a, b, d−g, 2c−f, 3c−g, 4a, c, e−g, 5a−e, and 6a, b and Supplementary Figs. 1, 2, 3b, c, 4, 5, and 6 are provided as a Source Data file. The FabA sequences used in this study are under the accession numbers AIN31422, NP_460041, and EFS15200. The genome sequences used in this study are under the accession numbers CP009273, NC_003197, NC_002516, NC_011283, NC_013716, CP001235, NC_004741, CP001589, and NC_009778. Other data supporting the findings of this study are available from the corresponding authors upon request.

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

## Acknowledgements

We thank Dr. Roy Curtiss III (University of Florida) for pRE112 and *E. coli* χ7213, Dr. Lixue Zhang (Qingdao University) for help on proteoliposome reconstitution, Dr. Qinggang Wang (Qingdao Institute of Bioenergy and Bioprocess Technology) for helpful discussion, and NBRP-E.coli at NIG for BW25113 strain. This study was

financially supported by the NSFC (31722001 and 31670089), and Natural Science Foundation of Shandong Province (JQ201707).

## Author contributions

G.Z. designed the experiments. Y.X., Z.Z., W.T., Y.S., Y.D., B.L., J.W., M.L., Y.W. and S.S. performed the experiments. G.Z., M.X., Y.X. and Q.Q. analyzed the results. G.Z., M.X., Y.X. and Q.Q. wrote the manuscript. All authors edited the manuscript before submission.

## Competing interests

The authors declare no competing interests.
