## [Peer Review File · Nature Communications]

Reviewers' comments:

Reviewer #1 (Remarks to the Author):

This is an interesting study and the authors have used a very impressive range of techniques in their study. Overall I think the conclusions they have reached are sound and represent an important finding that will be of significant interest in the field. However there are quite a lot of points that I would like them to correct or comment on first.

1. line 10 and many other places in the MS: It is not the case that all the AR systems known to date in E coli only function in stationary phase cells. For example, two of the reviews cited in this paper (Castanie-Cornet et al., 1999, and Foster, 2004) explicitly discuss the induction of AR2 in exponential phase cultures. The effectiveness of AR3 in exponential phase was demonstrated in Richard and Foster (2004). Constitutive expression of AR2 gives high levels of exponential phase acid resistance (Johnson et al., 2014). So the division of acid resistance systems into type I and II as proposed by the authors is very debatable. Many changes to the MS will be needed to address this point.

2. line 41: the presence of extracellular amino-acid is not always a requirement for induction of the AR2 - AR5 systems: AR2 for example is induced at pH 5.5 even in the absence of glutamate and glutamine (Sen et al, 2017).

3. Line 43 The function of AR2 is more complex than described here. The activity of the GadA/B/C proteins is key but the proteins whose expression is induced as part of the AR2 regulon include periplasmic chaperones HdeA and HdeD and several other proteins with important if not fully understood roles in acid resistance.

4. line 64 "enabled" rather than "conferred"

5. Figure 1 and in many other places throughout the paper. The authors use an unusual measure of what they call "growth" - the cfu ratio. As is very apparent from figure 1A this does not measure growth (although they describe it as growth at many points in the MS), but is a measure which combines attributes of both growth and survival (very different properties). In this figure the decline in CFU ratio is clearly caused as much by loss of viability of the cells at pH 4.2 as it is by continued growth at pH 7. I'm not clear why they use this measure, when showing growth curves as in figure 1A would have been more informative. In figure 1F for example they show the cfu ratio after 1 hour, but in figure 1G they show it after 30 minutes. Why were these times chosen, when as shown in figure 1A the differences are much clearer at later time points? This makes comparisons between figures difficult. Moreover, using this measure (as it is a ratio) makes it impossible to tell whether the differences seen between strains are caused by more rapid death or more rapid growth of the individual strains. This point needs addressing at multiple places in the paper. They sometimes even use the cfu ration to measure "growth rate" - it most definitely does not!

6. Also in Figure 1, it is stated that the cells are in "mid-log" phase when acidified. This is not the case: they are just about to enter early stationary phase, or at best "late log phase", as the gradient of the growth curve at this point (which is, correctly, plotted on a semi-log scale) is clearly beginning to deviate from a straight line. Expressions like mid-log and late-log are however not very accurate and should be avoided if possible. see for example the discussion and papers referred to here: <https://schaechter.asmblog.org/schaechter/2018/07/why-you-must-plot-your-growth-data-on-semi-log-graph-paper.html>

7. Lines 133-134 - how did they "analyse the promoter regions thoroughly"? The method used needs to be explained.

8. In figure 2 and elsewhere, the authors make extensive use of RT-PCR to establish the up-regulation of various genes. But the data is always presented as single bands, with no markers and no indication as to whether the PCRs were run on the same gels, and no indication of how reproducible the data were. RT-PCR per se is poorly quantitative because of the inherent problems of working with an amplification based system: this why qRT-PCR requires such care. They state in the methods that they have quantified their results (line 539), but these quantified results are not presented. They need to be, and it needs to be confirmed that the results have been gained under conditions where the level of the amplified fragment genuinely reflects the amount of starting mRNA. I refer the authors to the excellent biorad guide on this topic (http://www.bio-rad.com/webroot/web/pdf/lsr/literature/Bulletin_5279.pdf) - there are many others online.

On the topic of quantitation, the authors also make extensive use of Western blots to support their arguments. These are again notoriously difficult to quantify well. Ideally they should be repeated several times with standards at several dilutions to show they are in a linear range. The authors need to discuss how quantitative their data really are and how this may affect their conclusions.

9. The authors conclude that the CpxAR TCS is the key system involved in the response that they see, and I think their data is for the most part convincing. However, their discussion of CpxAR is rather limited. In particular, they fail to cite another paper that has already shown this TCS to have a significant role in the acid stress response (Surmann et al *Microbiology Open*. 2016 Aug;5(4):582-96). Surprisingly this paper shows that CpxAR strongly represses genes of the AR2 system, and that a CpxAR KO shows higher acid resistance. How does this fit with the story they are telling here? I think they need to make it clearer that CpxAR has a large and complex regulon (it interacts with the OmpR system, which itself plays a key role in acid resistance; see for example Stincone et al *Nucleic Acids Res*. 2011 Sep 1;39(17):7512-28. and Chakroborty and Kenney *Front Microbiol*. 2018 Nov 22;9:2656). This does not invalidate their central hypothesis but does give more of the context. Figure S7 implies that what they have found is the only relevant part of the system.

10. lines 194 - 199. what happens to the acid resistance phenotypes of the mutants where loss of cpxR binding in fabA or fabB has been engineered? These mutants provide a good way to test their central hypothesis.

11. Figure 3 and supporting text. I like the vesicle experiments very much. However it needs to be made clearer in the text that this is effectively an inverted system where the interior of the proteoliposomes corresponds to the periplasm (this can be seen in Figure S3 which will not be part of the main body of the paper).

12, line 299 - I don't understand and can't tell from the figure what the authors mean by "were distinguished by greyscale". Needs clarification. I think there are three different sets of results here which were rather variable (not unusual for animal studies). How were the p values calculated, and has any FDR been applied?

13 line 348. "Widespread" rather than "universal".

14. lines 451-453. The authors did not look at cyclopropane, but their phrasing implies that they did, but did not detect a change. Rephrase. (eg to "...we did not analyse CFA levels in this study")

15. Some methods are described too briefly or without supporting citations - see for eg lines 491 and 493.

Reviewer #2 (Remarks to the Author):

The manuscript entitled "A Novel Acid Resistant System Protecting Exponentially growing *Escherichia coli*" described the molecular regulation of a system that senses pH 4.2 through CpxRA and that, primarily through the fabA gene product, affects membrane fluidity in *E. coli*.

1) According to the title (and the text) the authors claim that they have identified a new AR system and also give to it a name Type II AR, unfortunately this is not correct: first of all because the system they are describing is not an Acid Resistance (AR) system rather it is an Acid Tolerance Response (ATR) system and secondly because it is not novel. Indeed Lin et al (J Bacteriol, 1995, 177(14): 4097–4104) described an ATR activated in exponential phase at pH 4.3 in minimal medium. What is new in this work is the characterization of the fabA expression by CpxRA.

2) I suggest to the authors to read and mention in the ms relevant review literature by Foster (J Microbiol, 2001, 39(2): 89-94) and Lund et al (FEMS Microbiol Rev, 2014, doi: 10.1111/1574-6976.12076) where not only the difference between AR and ATR is clearly explained, but also the role of UFAs as a mechanism in protection from acid stress is highlighted.

3) All the work in this ms stems from the observation that "E. coli improves growth status under acidic conditions by elevating the biosynthesis of unsaturated fatty acids" (lines 93-94). However the CFU ratio shown in Fig.1A is in log scale, whereas in Figs. 1F, 1G, 2D and 5A is in linear scale. so much smaller in comparison! This creates a lot of confusion and it is very difficult to appreciate the real relevance of findings.

4) What comes after in the rest of the ms is the deepening of the regulation of fadA gene via CpxRA and NlpE overexpression (OE). However, in this respect the work is quite confusing because many experiments are carried out with mutants created by insertion of a suicide vector (CpXR) and many other experiments are carried out overexpressing a gene of interest. The genetic backgrounds are therefore different and not always the intention and the results are directly comparable.

5) Fig1B and 2E seem to indicate that the only fatty acids in *E. coli* are 16:0, 16:1, 18:0 and 18:1, which should not be the case. Again reference and literature are missing and the authors should be more precise on this.

6) There are no error bars for pH 4.2 samples in Fig. 1B. Same in Fig. 2E in the wt pnlpE strain. In the latter panel it is not clear at which pH are carried out the measurement.

7) In Fig 2H the whole EMSA gel should be shown, or at least the wells (top part) should be visible

8) data in Fig 3F should show the results at pH 7.0: being Lys and Arg always positively charged (regardless the pH 4.2 or 7.0) I would expect that activation of FabA to be constitutive. This is also why I do not fully agree with the part of the sentence at lines 285-286.

9) In Fig 4F what is the pH of the culture without pH adjustments??

10) the section on bioproduction is not clear: it is not stated what is the pH of the culture when the fadA gene is expressed. A growth curve is not shown and the pH during growth was not monitored.....too little information to be meaningful.

11) Fig. 5B shows that the gene mostly affected by pH 5.0 is fabB but the authors show only the sequence of fabA

12) in the Discussion ATR should replace AR

Page 1: Title "Resistant" should be "Resistance". The title should be changed .

Page 3: Introduction – line 36: after "survival" can be suitably included the Ref FEMS Microbiol Rev, 2014, doi: 10.1111/1574-6976.12076 or <http://dx.doi.org/10.1016/bs.aambs.2015.03.002> where the *E. coli* AR systems are nicely described. However please note that these references are also OK when talking about ATR, which is the case in this ms.

Page 3: line 46 – other references such as Richard and Foster (2004), De Biase et al. (1999) as well as Castanie Cornet et al (1999) are also relevant here.

Page 4: line 71 – “Acid resistance” is not correct. You characterised an ATR.

Page 4: line 74– “something wrong with the sentence starting “when grown...”) The whole manuscript needs English editing

Page 8: lines 128-130 – The sentence starting from “Therefore.....” should be deleted

Page 9: Fig 2 – a diagram showing the genomic location and distance between fabA and fabB should be shown. Space can be saved by fusing 2A with 2 I and J.

Page 9: Fig2A and 2I and j can be fused

Page 9: Fig. 2I - wrong assignment of -35 in the fadA regulatory region . According to my inspection the “tagacg” is the most likely -35.

Page 9: Fig 2F – should include the test at pH 7.0 “-” the CpxR site. Now not shown.

Page 13: Fig 3A and B – why two panels and not just one?

Page 13: Fig 3D – please show also a lane with NlpE OE at pH 4.2

Page 15: lines 261-263: Based on the results shown in Fig 3, the sentence starting with “These results....” does not sound quite right.

Page 15: line 268-269 - include Ref FEMS Microbiol Rev , 2014, doi: 10.1111/1574-6976.12076 where the key role of His residues is also discussed

Page 16: The whole Fig 4 should be moved after the text line 306

Page 18: line 317 and Fig. 4B. I suggest to delete the figure or “data not shown” or “supplementary”

Page 18: line 325: delete “pretty”

Reviewer #3 (Remarks to the Author):

In the present manuscript, the authors characterized a new AR system where the two component system CpxRA directly senses acidification, and its regulatory circuit occurs during the exponential growing *E. coli* phase, as well as also occurs in other pathogenic bacterial.

The manuscript is written in a correct form, even when poor writing of some paragraphs disturbs the flow of ideas. The data showed represent advances in this field. The main concern when reading the manuscript is the absence of numerous controls in the assays, making that the manuscript requires major modifications before its acceptance.

Each questionable point is detailed below.

Major revision

Line 54-55: The author mentioned that “there is no report presenting *E. coli* growth at pH 4.5 or lower” However, numerous groups have investigated the growth of *E. coli* at different pH. See (Pienaar et al. 2018-Microbial Pathogenesis; De Biase and Peter A. 2015- Advances in Applied Microbiology; among others). Here the published results of DiGiuseppe and Silhavy (2003) Journal of Bacteriology, should be discussed. Authors should review and modify this sentence.

Line 78-79: The authors should analyze and show what happens if the bacteria is grown directly at pH 4.2, or pre-adapted to pH 5.

Line 81-82: The authors should mention the advantage of using this ration and explain what does it mean “growth state at acidic pH”.

Fig. 1 A and B: The error bars should be added.

Line 97: The authors should mention how much exposure time to pH 4.2 the bacteria were espoused. The authors must mention with respect to which they take out the % of each component.

Lines 100 and 102: The authors must modify the legend of Fig 1D and E, since the results were also determined at pH 7.

Line 135 (Fig 2A): To demonstrate that there is a consensus binding box of the CpxR into the promoters of the genes under study, the alignment must be performed with a consensus sequence of the regulator or with the binding sites identified in other CpxR-dependent genes, not only

between genes under study.

Line 137: Here the authors cannot conclude that CpxRA regulates the fab genes expression by simply identifying a possible binding site for CpxR regulator. The results from EMSA and S1 mapping assay should be discussed. This phrase must be modified.

Line 143: FabB is not "totally abolished by cpxR deletion. These results should be better discussed.

Line 145 to 147: In order to reach this conclusion, the authors must carry out the pH 4.2 resistance assay and the UFAs content in the cpxR mutant, without nlpE overexpression.

Fig 2F and G: It is not well understood what the represent data shown. What does CpxR site + or - mean? These binding sites were deleted from the promoter region of each gene?

Fig 2: Change the place of panel F, the reading of the figure becomes very confusing. The number of graphics presented in this figure is very large, and therefore should be divided into two figures to avoid confusion.

Fig 2, Line 152 and 158: The effect of cpxR deletion should be analyzed. Also other CpxRA system activation condition should be assayed.

Line 185: The authors must change the order of the graphs F, G, H, I and J in figure 2, since in the writing it is not possible to mention Fig 2I or 2J without having discussed the results of Fig 2F, 2G or 2H. This Figure can be divided into 2 or even 3 different figures, arranged according to the data and titles that appear in results.

Line 191 to 193: The authors must mention the result obtained that led them to this conclusion. The figure S1 should be modified for a better interpretation of the results, for example changing the + and - signs to wild type and double mutant as appropriate.

Line 196 to 199: This phrase must be modified to better understand in which region of the chromosome the CpxR site substitution is ... The authors will have made a substitution in the promoter region of each gene under study, where the possible binding site of the regulator is located?

In addition, the type of substitution should be mentioned.

Line 209: Authors should add a load control to demonstrate that the same amount was deposited on all wells and that the same number of cells was processed.

The Materials and Methods Section should be more specific. You should better explain the procedure carried out for the purification of these proteins, how the number of cells was normalized, how the phosphorylation of the regulator was obtained, the controls performed, etc.

Line 211: The authors should explain better how the phosphorylated form of CpxA was obtained, and its non-phosphorylated form in order to compare it in the gel.

These results are contradictory to those shown in 3A, where an increase in CpxA by pH 4.2 is detected. However, in 3B the levels of the unphosphorylated form are equal under all the pH analyzed.

Line 216 to 217: Apparently there is some error in the technique since if CpxA is being phosphorylated within proteoliposomes at pH 4.2, the unphosphorylated form should be of less intensity than that observed in proteoliposomes at pH 7 or 5.2, even less compared to the phosphorylated form at pH 4.2. This is not seen in Figure 3C.

According to those exposed in Materials and Methods, in the phosphorylation assay the proteoliposomes were incubated in a buffer at pH 7.5, the authors must explain what happens with the pH inside of the proteoliposomes and what changes the sensor could undergo to produce that the Kinase domain is phosphorylated or not.

Line 219 Fig 3C: In the assay many controls are missing, for example those that demonstrate that CpxR is actually phosphorylated by CpxA, these assays can be carried out with proteoliposomes incubated without the sensor.

Lines 234 and 238: Remove the phrases "See also Fig. S3 for illustration for the proteoliposome system" and "See also Fig. S4 for cartoon illustrating the PhoQ-CpxA fusion protein" from figure legend and put it into the results text.

Line 286, Fig 3G: In figure 3G a model of activation of the CpxAR system is proposed, this must be explained in this part of results or take out from the figure the 3G panel and place into the discussion section.

Line 287 to 288: Place Fig 4 below, after presenting the results plotted on it.

Line 310: Add "as control" after "vector". Change "and" by "or", between "pfabA, and pfabB".

Delete respectively, since the cell carries just one plasmid.

Line 319 to 321: The authors should review this phrase, something is missing, to understand that they analyzed the activity of ATPase before mentioning the results obtained.

Line 332 to 334: This assay should be better explained for a better interpretation, explain mainly why of 18 mice treated with the wt strain only 8 were examined, and why only 2 were challenged with the mutant. Check also this Mat and Met section.

Line 335: To make a correct statistical analysis the same number of animals must be considered with both variants...explain.

Line 345: The figure 4G does not exist, change by 4F

Line 351: What is the genetic source of this gene? Explain if in these two organisms this gene also exists. What is the percentage of identity that they do with this in E coli? These data should be here discussed.

Line 365: The authors must explain how they determined the ration, what was the control. The control without vector should be also added.

Line 369: Explain why was the test not performed at pH 4.2 as in E coli.

Line 385: The S7 scheme should be better explained.

Line 405 to 406: The authors are very conclusive in this sentence, so they should mention why they rule out that there is another activation pathway involved in this mechanism.

Line 576 to 577: Authors must specify when the CpxA sensor was added to the mix. They must also clarify whether the 300 μ mol of ATP enters or not into the lumen of the proteoliposomes, to allow the phosphorylation of the sensor, whose kinase site is on the outer side.

Line 580: How long was it incubated? Samples should have been taken at different times to see how the phosphorylated regulator band increases as the sensor band decreases.

Line 584: The authors should better explain the technique since although they reveal the gel with phosphorylated acrylamide and Mn²⁺ to give specificity for the phosphorylated form of the regulatory system's proteins, both forms are observed in Fig. 3 A, B and C, phosphorylated and not.

Line 595: Add reference after "calculated as previously described."

Line 623 to 624: According to results section, the number of mice used was not equal for each bacterial strain tested, it must be explained why. In addition, it should be explained why not all mice of the assay were used to recover the wt strain (only 8 of 18) and for the mutant (2 of 18).

Fig S1: Authors should modify the form used to represent results. For example in the legend of the figure: when mentioning wild type place sign - / + when the strain contains the empty vector, + / + when it carries pnlpE; or +/- when it is the double mutant containing pnlpE

Minor revision

Line 9: A space should be added between pH and 4.5

Line 17: and along the manuscript space should be added between pH and 4.5 or 7

Line 93: Add a point and leave space between Fig 1 and E. coli

Line 99: Delete "," between "synthase, play"

Line 103: Delete "s" from BW25113 strains

Line 104: Change "and" by "or" from "pfabA and pfabB"

Line 106 and 226: Unify "at pH4.2 at 30°C or 42°C for" and also "mL" writing on these lines and throughout the manuscript

Line 118: Delete "respectively" from "wild-type strain, respectively".

Line 352: Pass "respectively" after 2.07 fold

Line 477: Add the manufacturer name

Line 572: Add space between "20min"

Ref : NCOMMS-19-18242

A Novel Acid Tolerance Response System Protecting Exponentially Growing Escherichia coli

Response to the reviewers' comments:

We thank the Reviewers for their constructive comments and appreciation for the importance of our study: "*This is an interesting study and the authors have used a very impressive range of techniques in their study. Overall I think the conclusions they have reached are sound and represent an important finding that will be of significant interest in the field.*" (Reviewer 1); "*The manuscript..... described the molecular regulation of a system that senses pH 4.2 through CpxRA and that, primarily through the fabA gene product, affects membrane fluidity in E. coli.*" (Reviewer 2); "*The data showed represent advances in this field.*" (Reviewer 3).

We responded to Reviewers' comments in a point-by-point form.

We appreciate the opportunity to re-submit our study to Nature Communications and are happy to make any additional modifications suggested by the Editors and Reviewers.

Reviewer #1

1. line 10 and many other places in the MS: It is not the case that all the AR systems known to date in E coli only function in stationary phase cells. For example, two of the reviews cited in this paper (Castanie-Cornet et al., 1999, and Foster, 2004) explicitly discuss the induction of AR2 in exponential phase cultures. The effectiveness of AR3 in exponential phase was demonstrated in Richard and Foster (2004). Constitutive expression of AR2 gives high levels of exponential phase acid resistance (Johnson et al., 2014). So the division of acid resistance systems into type I and II as

proposed by the authors is very debatable. Many changes to the MS will be needed to address this point.

Reply: Thanks a lot for this correction. In the revised manuscript, we introduced that AR2 and AR3 can function during the exponential phase (page 3-4, line 49-51), and the definition of Type I and II acid resistance system was no longer used. The whole manuscript has been modified accordingly.

2. line 41: the presence of extracellular amino-acid is not always a requirement for induction of the AR2 - AR5 systems: AR2 for example is induced at pH 5.5 even in the absence of glutamate and glutamine (Sen et al, 2017).

Reply: we have rephrased this sentence to clarify that AR2 can be induced at acidic pH without glutamate, and appropriate reference was cited (page 3, line 42-45).

3. Line 43 The function of AR2 is more complex than described here. The activity of the GadaA/B/C proteins is key but the proteins whose expression is induced as part of the AR2 regulon include periplasmic chaperones HdeA and HdeD and several other proteins with important if not fully understood roles in acid resistance.

Reply: We have added more detail information about AR2 system and its regulatory network to show that AR2 is a system with high level of complexity in the revised manuscript (page 4, line 52-62).

4. line 64 "enabled" rather than "conferred"

Reply: It has been corrected in the revised manuscript (page 5, line 86).

5. Figure 1 and in many other places throughout the paper. The authors use an

unusual measure of what they call "growth" - the cfu ratio. As is very apparent from figure 1A this does not measure growth (although they describe it as growth at many points in the MS), but is a measure which combines attributes of both growth and survival (very different properties). In this figure the decline in CFU ratio is clearly caused as much by loss of viability of the cells at pH 4.2 as it is by continued growth at pH 7. I'm not clear why they use this measure, when showing growth curves as in figure 1A would have been more informative. In figure 1F for example they show the cfu ratio after 1 hour, but in figure 1G they show it after 30 minutes. Why were these times chosen, when as shown in figure 1A the differences are much clearer at later time points? This makes comparisons between figures difficult. Moreover, using this measure (as it is a ratio) makes it impossible to tell whether the differences seen between strains are caused by more rapid death or more rapid growth of the individual strains. This point needs addressing at multiple places in the paper. They sometimes even use the cfu ration to measure "growth rate" - it most definitely does not!

Reply: In this study, we used the value of CFU ratio (pH4.2/pH7.0) to measure the acid tolerance of exponentially growing E. coli, and explained why we use the CFU ratio instead of survival which was widely used to measure the bacterial acid resistance (page 6, line 104-114). In previous studies, survival was used to measure the bacterial acid resistance. Survival is calculated by the formula, survival = CFU after acid challenge/CFU before acid challenge × 100%, and the value of survival is usually much lower than 1. However, survival was determined under conditions E. coli can only survive without growth (pH 1.5-3.5), and is not suitable here as E. coli grew rapidly at pH 4.2 when CpxRA was activated or fab genes were overexpressed. The cell density after acid challenge will be much higher than that before acid challenge. So, the CFU ratio (pH 4.2/pH 7.0) is calculated to represent the acid tolerance of exponentially growing E. coli because growth of E. coli at pH 7.0 is steady. For a certain strain growing under certain conditions, the change of CFU ratio is

caused by the change of growth at pH 4.2. A value of CFU ratio close to 1 indicates that E. coli grows at pH 4.2 to a cell density similar to that at pH 7.0.

In this study, we challenged E. coli cells at pH 4.2 for 1 h or 30 min because under these conditions a moderate value of CFU ratio was obtained. So, we can see an increase of CUF ratio when CpxRA was activated or fab genes were overexpressed, and also can see a decrease of CFU ratio when cpxR or fab gene was mutated or the CpxR binding site was substituted. At later time points, it is hard to show the difference between wild-type strain and those mutant strains.

6. Also in Figure 1, it is stated that the cells are in "mid-log" phase when acidified. This is not the case: they are just about to enter early stationary phase, or at best "late log phase", as the gradient of the growth curve at this point (which is, correctly, plotted on a semi-log scale) is clearly beginning to deviate from a straight line. Expressions like mid-log and late-log are however not very accurate and should be avoided if possible. see for example the discussion and papers referred to here: <https://schaechter.asmblog.org/schaechter/2018/07/why-you-must-plot-your-growth-data-on-semi-log-graph-paper.html>

Reply: In our experiment, E. coli BW25113 strain was grown in medium E at pH 7.0 to a cell density of 3×10^8 CFU/mL. Then the cells were collected by centrifugation, washed and transferred into medium E at pH 7.0 and 4.2. We thought that is why the gradient of the growth curve changed at that time, because the growth curve of cells growing at pH 7.0 continuously is a straight line. For more accurate expression, we only mentioned the cell density, and didn't use mid-log or late-log anymore in the revised manuscript (page 6, line 96-99).

7. Lines 133-134 - how did they "analyse the promoter regions thoroughly"? The

method used needs to be explained.

Reply: We performed the alignment of DNA sequences of fabA and fabB promoter regions, and found a conserved direct repeat sequence GTAAA-(5 nt)-GCAAA, which is similar to the previously identified CpxR binding site. This part has been rephrased in the revised manuscript (page 8, line 145-148).

8. In figure 2 and elsewhere, the authors make extensive use of RT-PCR to establish the up-regulation of various genes. But the data is always presented as single bands, with no markers and no indication as to whether the PCRs were run on the same gels, and no indication of how reproducible the data were. RT-PCR per se is poorly quantitative because of the inherent problems of working with an amplification based system: this why qRT-PCR requires such care. They state in the methods that they have quantified their results (line 539), but these quantified results are not presented. They need to be, and it needs to be confirmed that the results have been gained under conditions where the level of the amplified fragment genuinely reflects the amount of starting mRNA. I refer the authors to the excellent biorad guide on this topic (http://www.bio-rad.com/webroot/web/pdf/lsr/literature/Bulletin_5279.pdf) - there are many others online.

Reply: Because of your concern about the repeatability and accuracy of RT-PCR results, we redo all these experiments using qRT-PCR. To ensure the repeatability, mRNA was isolated from at least two independent biological samples, and three technical repeats were performed for each biological sample. The expression of a reference gene, rpoD, was used as the normalizer to control for any difference in cDNA input amount. The relative difference of mRNA level was calculated using the $\Delta\Delta C_t$ method. The qRT-PCR results confirmed the upregulation of fab genes by acidic pH via CpxRA system (Fig 1e, 2c, 3d, 6b, and Supplementary Fig. 3).

On the topic of quantitation, the authors also make extensive use of Western blots to support their arguments. These are again notoriously difficult to quantify well. Ideally they should be repeated several times with standards at several dilutions to show they are in a linear range. The authors need to discuss how quantitative their data really are and how this may affect their conclusions.

Reply: In the revised manuscript, a representative result of western blot was shown in figures. Furthermore, quantitative data from at least two independent western blot experiments determined using ImageJ software (NIH) were presented, and loading controls were provided in the Source Data file. The results of multiple experiments are consistent, and the differences between various samples are statistically significant (Fig. 1d, 2b, 3c, 4a, 4c, 4e, 4f, and 4g). We believe that those results can support our conclusion.

9. The authors conclude that the CpxAR TCS is the key system involved in the response that they see, and I think their data is for the most part convincing. However, their discussion of CpxAR is rather limited. In particular, they fail to cite another paper that has already shown this TCS to have a significant role in the acid stress response (Surmann et al Microbiology Open. 2016 Aug;5(4):582-96). Surprisingly this paper shows that CpxAR strongly represses genes of the AR2 system, and that a CpxAR KO shows higher acid resistance. How does this fit with the story they are telling here? I think they need to make it clearer that CpxAR has a large and complex regulon (it interacts with the OmpR system, which itself plays a key role in acid resistance; see for example Stincone et al Nucleic Acids Res. 2011 Sep 1;39(17):7512-28. and Chakroborty and Kenney Front Microbiol. 2018 Nov 22;9:2656). This does not invalidate their central hypothesis but does give more of the context. Figure S7 implies that what they have found is the only relevant part of the system.

Reply: As your suggestion, we added that reference highlighting the important

role of CpxRA system in acid resistance (MicrobiologyOpen, 5: 582.), and the CpxRA-dependent gene regulation involved in exponential phase acid tolerance was discussed more thoroughly. Besides fab genes, CpxRA also activates the expression of genes related with cell wall modification, and thus increased the cell wall stability. It was reported that the log phase cells with overexpression of AR2 system genes are not more acid resistant (J Microbiol, 39: 89.). So, the CpxRA-mediated repression of AR2 system in log phase cells will guarantee that AR2 system is not induced in an inappropriate situation. Activation of fab genes and cell wall modification genes, repression of AR2 system by CpxRA, and crosstalking between CpxA and OmpR were discussed in the revised manuscript (page 20, line 420-440) and shown in Fig. 7.

10. lines 194 - 199. what happens to the acid resistance phenotypes of the mutants where loss of cpxR binding in fabA or fabB has been engineered? These mutants provide a good way to test their central hypothesis.

Reply: We have tested the acid tolerance of mutants with substitution in fabA or fabB CpxR binding site. As shown in Fig 3e, these mutants are more susceptible to acidic challenge than the wild-type strain (page 10, line 184).

11. Figure 3 and supporting text. I like the vesicle experiments very much. However it needs to be made clearer in the text that this is effectively an inverted system where the interior of the proteoliposomes corresponds to the periplasm (this can be seen in Figure S3 which will not be part of the main body of the paper).

Reply: We have rewritten this part to clarify that the CpxA protein is reconstituted into vesicles mainly in the inside-out orientation (page 11, line 210-212), and schematic diagram of the proteoliposome system (formerly Fig S3) was moved into the main body of the manuscript as Fig 4b.

12. line 299 - *I don't understand and can't tell from the figure what the authors mean by "were distinguished by greyscale". Needs clarification. I think there are three different sets of results here which were rather variable (not unusual for animal studies). How were the p values calculated, and has any FDR been applied?*

Reply: Sorry for the misleading information. In mouse survival passage experiment, three independent trials with six mice in each trail were performed for each strain. After 24 h, faecal samples from all 18 mice were collected to test the presence of each strain. Out of 18 inoculated mice, the wild-type BW25113 strain was positive in a total of 8 faecal samples, while the fabA CpxR site mutant was positive in only 2 faecal samples. We have rephased this sentence (page 14-15, line 290-295). The p value is calculated using Student's t-test.

13. line 348. *"Widespread" rather than "universal".*

Reply: It has been corrected in the revised manuscript (page 15, line 310).

14. lines 451-453. *The authors did not look at cyclopropane, but their phrasing implies that they did, but did not detect a change. Rephrase. (eg to "...we did not analyse CFA levels in this study")*

Reply: In this study, we analyzed the fatty acids content extracted from log phase E. coli cells by GC-MS, and CFA could not be detected (Fig. 1B), although it was regarded as a major factor in stationary phase acid resistance. However, UFA can be converted into CFA by CFA synthase, and increased UFAs content provides more substrate for the synthesis of CFA in stationary phase (page 20, line 402-406).

15. *Some methods are described too briefly or without supporting citations - see for eg lines 491 and 493.*

Reply: Some references were added, and the strain construction method was presented in detail (page 23, line 473-477).

Reviewer #2

1. *According to the title (and the text) the authors claim that they have identified a new AR system and also give to it a name Type II AR, unfortunately this is not correct: first of all because the system they are describing is not an Acid Resistance (AR) system rather it is an Acid Tolerance Response (ATR) system and secondly because it is not novel. Indeed Lin et al (J Bacteriol, 1995, 177(14): 4097–4104) described an ATR activated in exponential phase at pH 4.3 in minimal medium. What is new in this work is the characterization of the fabA expression by CpxRA.*

Reply: Thanks a lot for your suggestion. In the revised manuscript, we classified this CpxRA- and UFAs-dependent system as Acid Tolerance Response (ATR) system instead of the Acid Resistance (AR) system because it is towards moderate acid stress conditions. A brief overview of ATR systems activated at exponential phase and stationary phase and corresponding regulatory systems was provided (page 4, line 63-69). The definition of Type I and II acid resistance system was no longer used, and the whole manuscript has been modified accordingly.

2. *I suggest to the authors to read and mention in the ms relevant review literature by Foster (J Microbiol, 2001, 39(2): 89-94) and Lund et al (FEMS Microbiol Rev, 2014, doi: 10.1111/1574-6976.12076) where not only the difference between AR and ATR is clearly explained, but also the role of UFAs as a mechanism in protection from acid stress is highlighted.*

Reply: We have read carefully those two papers you mentioned and cited them in the revised manuscript. Both AR and ATR systems, along with their regulatory circuits, were introduced in the Introduction part (page 3-4, line 36-69). We made it clear that AR systems function under extreme acid stress and ATR systems are induced by moderate acid stress. The role of UFAs in bacterial acid tolerance was discussed (page 19-20, line 391-406). In our study, we found that the changes of UFAs contents in membrane lipid lower the membrane fluidity and F_0F_1 -ATPase activity, and improve the intracellular pH homeostasis and bacterial acid tolerance. Moreover, changes in fatty acid composition probably also affect the PTS system and enzyme secretion.

3. All the work in this ms stems from the observation that “E. coli improves growth status under acidic conditions by elevating the biosynthesis of unsaturated fatty acids” (lines 93-94). However the CFU ratio shown in Fig.1A is in log scale, whereas in Figs. 1F, 1G, 2D and 5A is in linear scale. so much smaller in comparison! This creates a lot of confusion and it is very difficult to appreciate the real relevance of findings.

Reply: Fig 1a has been revised to show the CFU ratio in linear scale to be easily compared with other results. In Fig. 1f, 2e, and 3e, the CFU ratio of BW25113 wild-type is 0.2-0.3 and determined after 1 h of exposure at pH 4.2. In Fig.1g, the CFU ratio is 0.5 and determined after 0.5 h acid challenge. Those are consistent with the results shown in Fig. 1a. All these results are calculated from three independent experiments, and the differences between various strains are statistically significant.

4. What comes after in the rest of the ms is the deepening of the regulation of fadA gene via CpxRA and NlpE overexpression (OE). However, in this respect the work is quite confusing because many experiments are carried out with mutants created by

insertion of a suicide vector (CpxR) and many other experiments are carried out overexpressing a gene of interest. The genetic backgrounds are therefore different and not always the intention and the results are directly comparable.

Reply: In this study, we proved that that the CpxRA system can be activated by acidic conditions, and regulates the transcription of fabA and fabB. Overexpression of NlpE is not required for the activation of fabA and fabB at acidic pH. As excessive NlpE protein was identified as a specific activating signal of CpxRA system (J Bacteriol, 1995, 177(4): 953-63; J Bacteriol, 2003, 185(8): 2432-40), we use NlpE to test whether the activated CpxRA can up-regulate fab genes and further change the membrane lipid composition of E. coli without acidic challenge, to conform the regulation of fab genes by CpxRA. In Fig. 1f-g, strain with overexpressed fab gene and strain with mutated fab gene were compared with wild-type BW25113 strain, demonstrating the important role of UFA synthesis in E. coli acid tolerance. In Fig. 2b-d, wild-type, activated CpxRA (by NlpE overexpression), and cpxR knockout mutant were compared to show the CpxRA-dependent regulation.

5. Fig1B and 2E seem to indicate that the only fatty acids in E. coli are 16:0, 16:1, 18:0 and 18:1, which should not be the case. Again reference and literature are missing and the authors should be more precise on this.

Reply: The reference was added in the revised manuscript (page 6-7, line 115-116). Phospholipids of E. coli cells were extracted and analyzed by GC-MS. The fatty acids were identified by comparing the retention times and mass fragmentation patterns with standards, and content of each fatty acid is given as the relative peak area [(peak area of one fatty acid/total peak area)×100%]. We have detected five fatty acids including C14:0, C16:0, C16:1, C18:0, and C18:1. These five fatty acids represent more than 96% of total fatty acids in E. coli cells, consistent with those results reported previously (Int J

Food Microbiol, 1997, 37: 163-173; Arch Microbiol, 1991, 157: 49-53.)

6. There are no error bars for pH 4.2 samples in Fig. 1B. Same in Fig. 2E in the wt pnlpE strain. In the latter panel it is not clear at which pH are carried out the measurement.

Reply: Error bars have been added in Fig 1b and Fig 2d. The experiment presented in Fig 2d was carried out at pH 7.0 (page 8, line 156-159). As overexpression of NlpE was an identified activating signal of CpxRA system, we use it to test whether the activated CpxRA can up-regulate fab genes and further change the membrane lipid composition of E. coli without acidic challenge.

7. In Fig 2H the whole EMSA gel should be shown, or at least the wells (top part) should be visible

Reply: Fig 2H (Fig 3f in the revised manuscript and Source Data file) was changed to show the whole EMSA gel. As the DNA fragments were labeled with ³²P, and signals were detected by autoradiography on X-ray film, the sample wells cannot be shown in this figure.

8. data in Fig 3F should show the results at pH 7.0: being Lys and Arg always positively charged (regardless the pH 4.2 or 7.0) I would expect that activation of FabA to be constitutive. This is also why I do not fully agree with the part of the sentence at lines 285-286.

Reply: According to your suggestion, we measured the protein level of FabA-his₆ at pH 7.0 in strains carrying wild-type CpxA, or CpxA variants including H52A, H52R, H52K, H117A, H117R and H117K. As shown in Supplementary Figure 6, the strain carrying CpxA variant with a basic amino

acid residue at position 52 or 117 presented a higher level of FabA protein at pH 7.0, when compared with strain with CpxA H52A or H117A mutant, probably due to the partial protonation of basic amino acid residue at pH 7.0 (page 13, line 257-261). So, we believed that the protonation of residues at positions 52 and 117 is enough to stimulate the regulatory activity of CpxRA.

9. In Fig 4F what is the pH of the culture without pH adjustments??

Reply: Without pH adjustments, the pH of culture was in the range of pH 5.2–5.5, and the pH of culture was added in Fig 5E in the revised manuscript.

10. the section on bioproduction is not clear: it is not stated what is the pH of the culture when the fabA gene is expressed. A growth curve is not shown and the pH during growth was not monitored.....too little information to be meaningful.

Reply: The fabA gene and genes for 3HP production are both under the control of an IPTG-inducible T7 promoter, and the expression was induced at an OD₆₀₀ of 0.8. Without fabA overexpression, the final cell density without pH adjustments was much lower than that with pH adjustments. When fabA is overexpressed, the OD₆₀₀ with and without pH adjustments were similar. In the revised manuscript, the final OD₆₀₀ and the pH of culture under different conditions were shown in Fig 5E.

11. Fig. 5B shows that the gene mostly affected by pH 5.0 is fabB, but the authors show only the sequence of fabA

Reply: The putative CpxR box sequences of fabB gene in various bacteria were added into Fig 6C in the revised manuscript.

12. *in the Discussion ATR should replace AR*

Reply: we used ATR to replace AR in the revised manuscript.

13. *Page 1: Title “Resistant” should be “Resistance”. The title should be changed .*

Reply: The title was changed to “A novel acid tolerance response system protecting exponentially growing *Escherichia coli*”.

14. *Page 3: Introduction – line 36: after “survival” can be suitably included the Ref FEMS Microbiol Rev , 2014, doi: 10.1111/1574-6976.12076 or <http://dx.doi.org/10.1016/bs.aambs.2015.03.002> where the *E. coli* AR systems are nicely described. However please note that these references are also OK when talking about ATR, which is the case in this ms.*

Reply: Thanks for your suggestion. In the revised manuscript, this paper was cited when introducing AR and ATR systems as well as their differences.

15. *Page 3: line 46 – other references such as Richard and Foster (2004), De Biase et al. (1999) as well as Castanie Cornet et al (1999) are also relevant here.*

Reply: The suggested references were added and discussed in the revised manuscript.

16. *Page 4: line 71 – “Acid resistance” is not correct. You characterised an ATR.*

Reply: In the revised manuscript, we classified this CpxRA- and UFAs-dependent system as Acid Tolerance Response (ATR) system instead of the Acid Resistance (AR) system because it is towards moderate acid stress conditions. The whole manuscript has been modified accordingly.

17. Page 4: line 74– “something wrong with the sentence starting “when grown...)
The whole manuscript needs English editing

Reply: This sentence has been rephrased (page 6, line 96-99).

18. Page 8: lines 128-130 – *The sentence starting from “Therefore.....” should be deleted*

Reply: It has been revised as suggested (page 8, line 141).

19. Page 9: Fig 2 – *a diagram showing the genomic location and distance between fabA and fabB should be shown. Space can be saved by fusing 2A with 2 I and J.*

Reply: A figure presenting the genomic locations of fab genes and corresponding CpxR sites was added as Fig 2a in the revised manuscript.

20. Page 9: *Fig2A and 2I and j can be fused*

Reply: It has been revised as suggested.

21. Page 9: *Fig. 2I - wrong assignment of -35 in the fabA regulatory region. According to my inspection the “tagacg” is the most likely -35.*

Reply: The -35 region of fabA was reassigned as suggested (Fig 3a).

22. Page 9: *Fig 2F – should include the test at pH 7.0 “-“ the CpxR site. Now not shown.*

Reply: The expression of fabA and fabB was tested in both protein and mRNA

level in BW25113 wild-type strain at pH 7.0 and 4.2, and strain with CpxR site substitution at pH 7.0 and 4.2, and shown as Fig 3c and 3d in the revised manuscript.

23. *Page 13: Fig 3A and B – why two panels and not just one?*

Reply: Fig 3A and 3B were combined into one panel as Fig 4a in the revised manuscript.

24. *Page 13: Fig 3D – please show also a lane with NlpE OE at pH 4.2*

Reply: In the revised manuscript, we showed the protein level of SlyB and RstA (PhoPQ-activated genes) with wild-type PhoQ and PhoQ-CpxA fusion. The experiments were performed under four conditions: pH 7.0, pH 4.2, pH 7.0 with NlpE overexpression, and pH 4.2 with NlpE overexpression. As shown in Fig. 4e, SlyB and RstA were activated via the wild-type PhoQ only by pH 4.2, and they could be up-regulated via the PhoQ-CpxA fusion by both pH 4.2 and NlpE overexpression (page 12, line 226-229).

25. *Page 15: lines 261-263: Based on the results shown in Fig 3, the sentence starting with “These results....” does not sound quite right.*

Reply: This sentence has been rephrased (page 12, line 235-237).

26. *Page 15: line 268-269 - include Ref FEMS Microbiol Rev, 2014, doi: 10.1111/1574-6976.12076 where the key role of His residues is also discussed*

Reply: It has been revised as suggested.

27. Page 16: The whole Fig 4 should be moved after the text line 306

Reply: It has been revised as suggested.

28. Page 18: line 317 and Fig. 4B. I suggest to delete the figure or “data not shown” or “supplementary”

Reply: This figure was shown as Supplementary Fig 7 in the revised manuscript.

29. Page 18: line 325: delete “pretty”

Reply: It has been revised as suggested (page 14, line 285).

Reviewer #3

Major revision

1. Line 54-55: The author mentioned that “there is no report presenting *E. coli* growth at pH 4.5 or lower” However, numerous groups have investigated the growth of *E. coli* at different pH. See (Pienaar et al. 2018-Microbial Pathogenesis; De Biase and Peter A. 2015- Advances in Applied Microbiology; among others). Here the published results of DiGiuseppe and Silhavy (2003) *Journal of Bacteriology*, should be discussed. Authors should review and modify this sentence.

Reply: Thanks for your suggestion. We have consulted a lot of literatures, and did not found that *E. coli* can grow below pH 4.0. In some references (J Bacterial, 1994, 177:4097; Appl Environ Microbiol, 2015, 81:1932), authors clarified that the acid limit for *E. coli* growth is pH 4.5. The results of Lin et al. (J Bacterial, 177:4097) showed that *E. coli* could grow at pH 4.3 but not at pH 4.2 in buffered LB medium. The results of Kaur et al. (J Cell Sci Ther, 2017, 8:1000260) showed that *E. coli* could grow at pH 4.5 but not at pH 4.0 in

minimal medium M9. In the paper you mentioned (Microb Pathogenesis, 2018, 128:396), it was presented that the cell density of *E. coli* did not increase during 3 h of exposure to pH 4.5. In this paper, there was a figure (see below), probably making reader think that *E. coli* could grow at pH 2.5-4.5.

Fig. 5. Recovery (proliferation) of EPEC post SGF exposure over an 8 h period. Cells grown in nutrient broth were used as positive controls.

In this experiment, enteropathogenic *E. coli* (EPEC) was treated in simulated gastric fluid (SGF) at pH 1.5, 2.5, 3.5 and 4.5 for 0, 30, 60, 120 or 180 min. Then, the cells were collected, resuspended in Mueller Hinton II Broth (pH 7.3), and further incubated at 37 °C for 8 h with agitation. Cell recovery and proliferation was measured using optical density (OD) at 600 nm. These results demonstrated the capacity of *E. coli* to recover from acid exposure of pH 2.5-4.5, but not the capacity of *E. coli* to grow at pH 2.5-4.5.

2. Line 78-79: *The authors should analyze and show what happens if the bacteria is grown directly at pH 4.2, or pre-adapted to pH 5.*

Reply: We have tested the direct inoculation of *E. coli* into medium E at pH 4.2. BW25113 strain was grown overnight in LB medium at pH 7.0 or 5.0, and then diluted 1:50 into medium E at pH 4.2. However, the cells were rapidly killed even preadapted at pH 5.0. This result was shown in the revised manuscript as Supplementary Fig. 1 (page 5-6, line 92-96).

3. Line 81-82: The authors should mention the advantage of using this ration and explain what does it mean "growth state at acidic pH".

Reply: In this study, we used the value of CUF ratio (pH4.2/pH7.0) to measure the acid tolerance of exponentially growing *E. coli*, and explained why we use the CFU ratio instead of survival which was widely used to measure the bacterial acid resistance (page 6, line 104-114). In previous studies, survival was used to measure the bacterial acid resistance. Survival is calculated by the formula, survival = CFU after acid challenge/CFU before acid challenge × 100%, and the value of survival is usually much lower than 1. However, survival was determined under conditions *E. coli* can only survive without growth (pH 1.5-3.5), and is not suitable here as *E. coli* grew rapidly at pH 4.2 when CpxRA was activated or fab genes were overexpressed. The cell density after acid challenge will be much higher than that before acid challenge. So, the CFU ratio (pH 4.2/pH 7.0) is calculated to represent the acid tolerance of exponentially growing *E. coli* because growth of *E. coli* at pH 7.0 is steady. For a certain strain growing under certain conditions, the change of CFU ratio is caused by the change of growth at pH 4.2. A value of CFU ratio close to 1 indicates that *E. coli* grows at pH 4.2 to a cell density similar to that at pH 7.0.

4. Fig. 1 A and B: The error bars should be added.

Reply: Error bars were added in Fig 1a and 1b.

5. Line 97: *The authors should mention how much exposure time to pH 4.2 the bacteria were espoused. The authors must mention with respect to which they take out the % of each component.*

Reply: The cells were grown in medium E at pH 7.0 to a cell density of 3×10^8 CFU/mL, collected and transferred into the same medium at pH 7.0 and 4.2, respectively. After 1-h of exposure at different pH, phospholipids were extracted from E. coli cells, and analyzed by GC-MS. Content of each fatty acid is given as the relative peak area [(peak area of one fatty acid/total peak area) \times 100%]. The above information was added into the revised manuscript (page 6, line 115-116; page 40, line 841-842).

6. Lines 100 and 102: *The authors must modify the legend of Fig 1D and E, since the results were also determined at pH 7.*

Reply: The legend of Fig 1d and 1e was revised accordingly (page 40, line 839-844).

7. Line 135 (Fig 2A): *To demonstrate that there is a consensus binding box of the CpxR into the promoters of the genes under study, the alignment must be performed with a consensus sequence of the regulator or with the binding sites identified in other CpxR-dependent genes, not only between genes under study.*

Reply: The alignment of CpxR binding site sequence and promoter sequences of fabA and fabB was carried out and shown as Supplementary Fig. 2 in the revised manuscript.

8. Line 137: *Here the authors cannot conclude that CpxRA regulates the fab genes*

expression by simply identifying a possible binding site for CpxR regulator. The results from EMSA and SI mapping assay should be discussed. This phrase must be modified.

Reply: We are trying to express that discovery of putative CpxR binding site raised the hypothesis that transcription of *fabA* and *fabB* is regulated by two-component system CpxRA. The part has been rephrased (page 8, line 148-151). The results of gel-shift and DNase I footprinting were discussed in the following (page 10, line 186-193).

9. Line 143: FabB is not "totally abolished by cpxR deletion. These results should be better discussed.

Reply: Here, we mean that the activating effect of NlpE overexpression was totally abolished by *cpxR* deletion, but not that the expression of *fabA* and *fabB* was totally abolished by *cpxR* deletion. To make it easier to understand, we rephrased that sentence (page 8, line 154-156).

10. Line 145 to 147: In order to reach this conclusion, the authors must carry out the pH 4.2 resistance assay and the UFAs content in the cpxR mutant, without nlpE overexpression.

Reply: To demonstrate the regulation of *fab* genes by CpxRA system, we want to test the expression of *fabA* and *fabB* under CpxRA activating conditions other than acidic pH. So, we cloned *nlpE* gene because overexpression of NlpE was identified as a specific signal activating CpxRA system (J Bacteriol, 1995, 177: 953-63; J Bacteriol, 2003, 185: 2432-40). As shown in Fig. 2b, 2c and 2d, excessive NlpE protein significantly enhanced the expression level of *fab* genes and UFAs content in membrane lipid at pH 7.0, and deletion of *cpxR* totally abolished this activating effect of NlpE. These results demonstrated that

activation of fab genes by NlpE overexpression is dependent on the CpxRA system, and the NlpE protein itself cannot stimulate the transcription of fab genes. Thus, overexpression of NlpE in cpxR mutant should not play any role. As predicted, the cpxR mutants carrying empty vector or nlpE expressing plasmid showed similar tolerance to acidic pH, much lower than that of BW25113 wild-type strain (Fig. 2e).

11. Fig 2F and G: It is not well understood what the represent data shown. What does CpxR site + or - mean? These binding sites were deleted from the promoter region of each gene?

Reply: + means the wild-type CpxR box, and – means substituted CpxR box. For better understanding, it has been modified in the revised manuscript as Fig 3c and 3d.

12. Fig 2: Change the place of panel F, the reading of the figure becomes very confusing. The number of graphics presented in this figure is very large, and therefore should be divided into two figures to avoid confusion.

Reply: This figure has been divided into Fig 2 and Fig3 in the revised manuscript. Fig 2 shows the results demonstrating that CpxRA regulates the transcription of fabA and fabB, and Fig 3 shows the results related with the identification of CpxR box in fabA and fabB promoter regions.

13. Fig 2, Line 152 and 158: The effect of cpxR deletion should be analyzed. Also other CpxRA system activation condition should be assayed.

Reply: In Fig 2e, we compared the CFU ratio of wild-type BW25113 strain and cpxR knockout mutant, demonstrating that the CpxR is essential for the acid tolerance of exponentially growing E. coli.

In this study, we proved that the CpxRA system can be activated by acidic conditions, and regulates the transcription of *fabA* and *fabB*. Overexpression of NlpE is not required for the activation of *fabA* and *fabB* at acidic pH. As excessive NlpE protein was identified as a specific activating signal of CpxRA system (J Bacteriol, 1995, 177: 953-63; J Bacteriol, 2003, 185: 2432-40), we use NlpE to test whether the activated CpxRA can up-regulate *fab* genes and further change the membrane lipid composition of *E. coli* without acidic challenge, to conform the regulation of *fab* genes by CpxRA.

14. Line 185: The authors must change the order of the graphs F, G, H, I and J in figure 2, since in the writing it is not possible to mention Fig 2I or 2J without having discussed the results of Fig 2F, 2G or 2H. This Figure can be divided into 2 or even 3 different figures, arranged according to the data and titles that appear in results.

Reply: This figure has been divided into Fig 2 and Fig 3 in the revised manuscript. Fig 2 shows the results demonstrating that CpxRA regulates the transcription of *fabA* and *fabB*, and Fig 3 shows the results related with the identification of CpxR box in *fabA* and *fabB* promoter regions.

15. Line 191 to 193: The authors must mention the result obtained that led them to this conclusion. The figure S1 should be modified for a better interpretation of the results, for example changing the + and - signs to wild type and double mutant as appropriate.

Reply: This sentence was rephrased as following: qRT-PCR result showed that the mRNA level of *fabA* and *fabB* was upregulated by NlpE overexpression in a *fabR fadR* double mutant strain, indicating that they had no obvious effect on CpxRA-dependent activation of *fabA* and *fabB* (page 9, line 172-175). This figure was also modified for a better interpretation (Supplementary Fig. 3 in the revised manuscript).

16. Line 196 to 199: This phrase must be modified to better understand in which region of the chromosome the CpxR site substitution is ... The authors will have made a substitution in the promoter region of each gene under study, where the possible binding site of the regulator is located? In addition, the type of substitution should be mentioned.

Reply: In the revised manuscript, we added a schematic diagram to show the genomic locations of fabA and fabB genes and corresponding CpxR sites (Fig 2a). In those CpxR site mutant strains, the CpxR box of fabA or fabB was replaced with sequence CATCT-(5 nt)-CATCT (page 9, line 178-181).

17. Line 209: Authors should add a load control to demonstrate that the same amount was deposited on all wells and that the same number of cells was processed. The Materials and Methods Section should be more specific. You should better explain the procedure carried out for the purification of these proteins, how the number of cells was normalized, how the phosphorylation of the regulator was obtained, the controls performed, etc.

Reply: To monitor the CpxA protein level, a strain carrying chromosomal His₆-tagged CpxA was grown in medium E at pH 7.0 to an OD₆₀₀ of 0.6. The cells were collected, washed and resuspended in medium E at pH 7.0, 5.2, and 4.2 for further growth of 1 h. Then the E. coli cells were harvested and disrupted by sonication, the cell lysates were centrifuged, and the supernatants were used for SDS-PAGE. Protein concentration was determined using BCA protein assay kit, and the same amount of total cellular protein were subjected to SDS-PAGE in duplicate gels, one for Coomassie blue staining as loading control, the other for western blot to detect the CpxA level. The method section was rephased (page 24, line 501-511), and loading control was provided in Source Data file.

18. Line 211: The authors should explain better how the phosphorylated form of CpxA was obtained, and its non-phosphorylated form in order to compare it in the gel. These results are contradictory to those shown in 3A, where an increase in CpxA by pH 4.2 is detected. However, in 3B the levels of the unphosphorylated form are equal under all the pH analyzed.

Reply: In the revised manuscript, we introduced how phosphorylation occurs and determines the regulatory activity of CpxRA. As a sensor histidine kinase, CpxA spans the cell membrane and exposes its sensor domain into the periplasm. When CpxA detects a specific signal, it autophosphorylates and then transports the phosphoryl group to its cognate regulator CpxR, enabling the regulatory activity of CpxR. Without inducing signal, CpxA acts as a phosphatase to maintain CpxR in an inactive state (page 10, line 197-202).

In Fig. 4a upper section in the revised manuscript, whole-cell lysates from E. coli cells treated at different pH were normalized to the same amount of total protein, and analyzed using western blot to determine the expression of CpxA (page 43, line 900-902). The result showed that acid shock significantly enhanced the CpxA protein level.

In Fig. 4a middle section in the revised manuscript, CpxA-His₆ protein were purified from E. coli cells treated at different pH using Ni-affinity chromatography as previously described (J Biol Chem, 2007, 282:8583), and the same amount of purified CpxA-His₆ protein was subjected to western blot analysis (page 43, line 902-905). So, we can only see the increased level of phosphorylated CpxA in this figure.

19. Line 216 to 217: Apparently there is some error in the technique since if CpxA is being phosphorylated within proteoliposomes at pH 4.2, the unphosphorylated form should be of less intensity than that observed in proteoliposomes at pH 7 or 5.2, even

less compared to the phosphorylated form at pH 4.2. This is not seen in Figure 3C.

According to those exposed in Materials and Methods, in the phosphorylation assay the proteoliposomes were incubated in a buffer at pH 7.5, the authors must explain what happens with the pH inside of the proteoliposomes and what changes the sensor could undergo to produce that the Kinase domain is phosphorylated or not.

Reply: As you said, the unphosphorylated proteins should be less with internal pH 4.2 as the phosphorylated forms increased. However, it is not seen (Fig. 4c in the revised manuscript). In this experiment, the amount of phosphorylated proteins was much lower when compared with that of unphosphorylated proteins. To detect the signal of phosphorylated protein, we had to increase the sample loading and prolong the film exposure time in western blot, which may lead to the overexposure of those unphosphorylated proteins. This could be the reason why we didn't see the decrease of unphosphorylated proteins.

The proteoliposome system can ensure the difference between internal and external pH, and has been used to characterize the pH-sensing mechanism (J Biol Chem, 2007, 282:8583; Proc Natl Acad Sci USA, 2008, 105:6900). We have clarified that in the revised manuscript (page 11, line 208-210). So, the pH inside vesicles will remain acidic when incubated in the phosphorylation buffer at pH 7.5.

20. Line 219 Fig 3C: In the assay many controls are missing, for example those that demonstrate that CpxR is actually phosphorylated by CpxA, these assays can be carried out with proteoliposomes incubated without the sensor.

Reply: As suggested, we carried out the proteoliposome experiment without the sensor kinase CpxA, and no phosphorylated CpxR protein was detected. This has been added in the revised manuscript (page 11, line 214-217).

21. Lines 234 and 238: Remove the phrases “See also Fig. S3 for illustration for the proteoliposome system” and “See also Fig. S4 for cartoon illustrating the PhoQ-CpxA fusion protein” from figure legend and put it into the results text.

Reply: For better interpretation, previous Fig. S3 and S4 were moved into the main body of manuscript as Fig. 4b and 4d. The figure legend was revised accordingly (page 43, line 907 and 914-918).

22. Line 286, Fig 3G: In figure 3G a model of activation of the CpxAR system is proposed, this must be explained in this part of results or take out from the figure the 3G panel and place into the discussion section.

Reply: The previous Fig. 3G was combined with Fig. 7 in the revised manuscript. The activation of CpxA by acidic pH via histidine protonation was discussed in the Discussion section (page 20, line 407-419).

23. Line 287 to 288: Place Fig 4 below, after presenting the results plotted on it.

Reply: It has been revised as suggested.

24. Line 310: Add "as control" after “vector”. Change "and" by "or", between “*pfabA*, and *pfabB*”. Delete respectively, since the cell carries just one plasmid.

Reply: It has been revised as suggested (page 14, line 270).

25. Line 319 to 321: The authors should review this phrase, something is missing, to understand that they analyzed the activity of ATPase before mentioning the results obtained.

Reply: This phrase was modified as following: F₀F₁-ATPase spans the cell

membrane and transports periplasmic protons to cytoplasm with production of ATP, and lipids are required for its optimal functioning. The activity of F₀F₁-ATPase was measured, suggesting that the enhancement of UFAs content repressed its activity. (page 14, line 280-282)

26. Line 332 to 334: This assay should be better explained for a better interpretation, explain mainly why of 18 mice treated with the wt strain only 8 were examined, and why only 2 were challenged with the mutant. Check also this Mat and Met section.

Reply: Sorry for the misleading information. In mouse survival passage experiment, three independent trials with six mice in each trial were performed for each strain. After 24 h, faecal samples from all 18 mice were collected to test the presence of each strain. Out of 18 inoculated mice, the wild-type BW25113 strain was positive in a total of 8 faecal samples, while the fabA CpxR site mutant was positive in only 2 faecal samples. We have rephased this sentence (page 14-15, line 290-295).

27. Line 335: To make a correct statistical analysis the same number of animals must be considered with both variants...explain.

Reply: As mentioned above, three independent trials with six mice in each trial were performed. So, a total of 18 mice were used for each strain.

28. Line 345: The figure 4G does not exist, change by 4F

Reply: it has been corrected as suggested (Fig 5e in the revised manuscript, page 15, line 304).

29. Line 351: What is the genetic source of this gene? Explain if in these two

organisms this gene also exists. What is the percentage of identity that they do with this in E coli? These data should be here discussed.

Reply: The genes *fabA* and *fabB* are present in both *Salmonella typhimurium* LT2 and *Shigella flexneri* 2a str. 2457T. As the FabA protein sequence of *E. coli* BW25113 shares more than 98% identity with those of *S. typhimurium* LT2 and *S. flexneri* 2a str. 2457T, *fabA* gene of *E. coli* BW25113 was cloned and transformed into those two strains. These data were added in the revised manuscript (page 15-16, line 312-314 and Supplementary Fig. 8).

30. Line 365: The authors must explain how they determined the ration, what was the control. The control without vector should be also added.

Reply: The *S. typhimurium* LT2 and *S. flexneri* 2a str. 2457T strains without the empty vector were also tested, and the corresponding data were added in the revised manuscript (page 16, line 317-320 and Fig. 6a). There is no significant difference between the strains with and without the empty vector, and the *fabA* overexpression elevated the CFU ratios by 2.46- and 2.07-fold, respectively, after acidic challenge at pH 5.0 for 1 h when compared with those strains carrying empty vector.

31. Line 369: Explain why was the test not performed at pH 4.2 as in E coli.

Reply: We have tried to challenge them at pH 4.2 for 1 h, but no viable cells can be detected, suggesting that *S. typhimurium* LT2 and *S. flexneri* 2a str. 2457T were more acid sensitive than *E. coli*. So we changed the condition to pH 5.0.

32. Line 385: The S7 scheme should be better explained.

Reply: The previous Fig. S7 was modified to include more CpxR-regulated genes involved in bacterial acid tolerance (Fig. 7 in the revised manuscript). Besides fab genes, CpxRA also activates the expression of genes related with cell wall modification, and thus increased the cell wall stability. The expression of AR2 systems genes was repressed by CpxRA system. As the log phase cells with overexpression of AR2 system genes are not more acid resistant (J Microbiol, 39: 89.), the CpxRA-mediated repression of AR2 system in log phase cells will guarantee that AR2 system is not induced in an inappropriate situation. Activation of CpxRA by acid pH, CpxRA-regulated genes involved in bacterial acid tolerance, and putative function of CpxRA in bacterial pathogenesis were discussed more thoroughly in the revised manuscript (page 20-22, line 407-452).

33. Line 405 to 406: The authors are very conclusive in this sentence, so they should mention why they rule out that there is another activation pathway involved in this mechanism.

Reply: As CpxA is capable of phosphorylating CpxR upon exposure to pH 4.2 in a reconstituted proteoliposome system (Fig. 4C), we believe that the UFAs-CpxRA system can be activated by acidic pH alone. This sentence has been rephrased (page 17-18, line 354-359).

34. Line 576 to 577: Authors must specify when the CpxA sensor was added to the mix. They must also clarify whether the 300 μ mol of ATP enters or not into the lumen of the proteoliposomes, to allow the phosphorylation of the sensor, whose kinase site is on the outer side.

Reply: The purified CpxA-His₆ protein was added into the mix before Bio-Beads treatment, which was clarified in the manuscript (page 26, line 542-543).

As a typical sensor kinase of two-component system, CpxA exposes its sensor domain into the periplasm, and the histidine kinase and phosphate transferase domain is in the cytoplasm. Once a specific signal is detected by the sensor domain, CpxA autophosphorylates and then transfers the phosphoryl group to cognate regulator CpxR. In proteoliposome, CpxA protein was reconstituted into vesicles mainly in the inside-out orientation (Fig. 4b). So, the phosphorylation happens outside the proteoliposome, and ATP is not required to enter the lumen of vesicles.

35. Line 580: How long was it incubated? Samples should have been taken at different times to see how the phosphorylated regulator band increases as the sensor band decreases.

Reply: To analyze phosphotransfer, purified His₆-CpxR was added to this mixture and incubated at room temperature for 20 min (page 26, line 551-553).

According to your suggestion, we carried out the phosphotransfer experiment with different incubation time. However, we could not observe difference in the amount of phosphorylated proteins.

To analyze the phosphorylated CpxA and CpxR, we have to separate the CpxR protein and CpxA-containing proteoliposome with ultracentrifugation of 30 min. We presumed that the phosphotransfer reaction continued during centrifugation, and that could be the reason why no difference was observed.

36. Line 584: The authors should better explain the technique since although they reveal the gel with phosphorylated acrylamide and Mn²⁺ to give specificity for the phosphorylated form of the regulatory system's proteins, both forms are observed in Fig. 3 A, B and C, phosphorylated and not.

Reply: The gel used to detect phosphorylated protein contains 50 μM

Phos-tagTM acrylamide (Wako Chemicals) and 100 μM Mn^{2+} . This gel is not specific for phosphorylated proteins, but Phos-tagTM acrylamide and Mn^{2+} can retard the mobility of phosphoproteins to show the phosphorylated and non-phosphorylated forms in two separated bands. So, both forms can be observed in Fig 4. We have rewritten this part (page 27, line 554-558).

37. Line 595: Add reference after “calculated as previously described.”

Reply: It has been corrected as suggested (page 28, line 592).

38. Line 623 to 624: According to results section, the number of mice used was not equal for each bacterial strain tested, it must be explained why. In addition, it should be explained why not all mice of the assay were used to recover the wt strain (only 8 of 18) and for the mutant (2 of 18).

Reply: Sorry for the misleading information. In mouse survival passage experiment, three independent trials with six mice in each trail were performed for each strain. After 24 h, faecal samples from all 18 mice were collected to test the presence of bacteria strain. Out of 18 inoculated mice, the wild-type BW25113 strain was positive in a total of 8 faecal samples, while the *fabA* CpxR site mutant was positive in only 2 faecal samples. We have rephased this sentence (page 14-15, line 290-295).

*39. Fig S1: Authors should modify the form used to represent results. For example in the legend of the figure: when mentioning wild type place sign - / + when the strain contains the empty vector, + / + when it carries *pnlpE*; or +/- when it is the double mutant containing *pnlpE**

Reply: This figure was modified for a better interpretation (Supplementary Fig 3 in the revised manuscript).

Minor revision

40. Line 9: A space should be added between pH and 4.5

Reply: This sentence was deleted in the revised manuscript. We checked the whole manuscript to make sure that there is a space between pH and number.

41. Line 17: and along the manuscript space should be added between pH and 4.5 or 7

Reply: It has been corrected as suggested. We checked the whole manuscript to make sure that there is a space between pH and number.

42. Line 93: Add a point and leave space between Fig 1 and E. coli

Reply: It has been corrected as suggested (page 40, line 829).

43. Line 99: Delete “,” between “synthase, play”

Reply: It has been corrected as suggested (page 40, line 839).

44. Line 103: Delete “s” from BW25113 strains

Reply: It has been corrected as suggested (page 40, line 844).

45. Line 104: Change “and” by “or” from “pfabA and pfabB”

Reply: It has been corrected as suggested (page 40, line 845).

46. Line 106 and 226: Unify “at pH4.2 at 30°C or 42°C for” and also “mL” writing on these lines and throughout the manuscript

Reply: It has been corrected as suggested throughout the manuscript.

47. Line 118: Delete “respectively” from “wild-type strain, respectively”.

Reply: It has been corrected as suggested (page 7, line 132).

48. Line 352: Pass “respectively” after 2.07 fold

Reply: It has been corrected as suggested (page 16, line 320).

49. Line 477: Add the manufacturer name

Reply: The manufacturer of Luria-Bertani broth was added (page 22, line 458), and the E minimal medium is not a commercial product.

50. Line 572: Add space between “20min”

Reply: It has been corrected as suggested (page 26, line 543).

REVIEWERS' COMMENTS:

Reviewer #1 (Remarks to the Author):

Thanks to the authors for the changes made to the manuscript. I think the paper is now much clearer. I have a few minor corrections that I suggest should still be made:

lines 105/106 should read = (CFU after acid challenge/CFU before acid challenge) x 100%

line 121 should read "as a consequence"

lines 209/210 would probably be better written as "where a different internal and external pH can be stably maintained"

lines 372 change to "is likely to play a key role"

line 396 change to "mutans"

line 406 needs rephrasing, I think to something like "will potentially enable the synthesis of..."

Peter Lund

Reviewer #2 (Remarks to the Author):

The ms has significantly improved since its first submission. However few points still need to be addressed, mostly to improve the clarity.

Page 1 , line 5 - Running Title: Replace "Acid Resistance" with "Acid Tolerance"

Page 2 – Abstract: two abbreviation (ATR and UFAs) are used, but reused only once. Maybe they could be removed in the abstract?

Page 2, line 9 – "Here, we characterized a new acid tolerance response (ATR) system and its regulatory circuit, required for E. coli growth at pH 4.2" should be replaced with "Here, we show for the first time the regulatory circuit controlling the acid tolerance response (ATR) system required for E. coli growth at pH 4.2" (see also the point immediately below)

Page 2, lines 18-19 – About the sentence starting with "This new ATR system is conserved...." because the occurrence of this system is NOT new (Lin et al., 1995 reported it) I suggest to rephrase "The molecular aspects of this ATR system, which is conserved across bacteria species such as Salmonella, Shigella, Klebsiella, Pseudomonas, are herein dissected for the first time"

Page 2, line 21 – replace the word "resistance" with "tolerance"

Page 2, line 24 – Keywords: don't think "Acid Resistance" is anymore necessary?

Page 2, line 26 – Keywords: "membrane lipid" should be "membrane fluidity" or "membrane lipid composition"

Page 5, line 73 – "nongrowth" should be "no-growth"?

Page 7, line 121 – After listing the different fatty acids I suggest to the authors to include the sentence that they used in the rebuttal to my comment “These five fatty acids represent more than 96% of total fatty acids in *E. coli* cells, consistent with those results reported previously (Int J Food Microbiol, 1997, 37: 163-173; Arch Microbiol, 1991, 157: 49-53.)”

Figure 2a – Along the line representing the chromosome of *E. coli* BW25113 the slash symbols “//” should be correctly placed before *fabA*, between *fabA* and *fabB* and after *fabB* gene to clearly show that the drawing is not “in scale”.

Page 13, line 261 – Remove "partial"

Page 13, lines 261-264 – Still not sure why the authors keep saying that the presence of the His residue I position 52 and 117 is “not a specific requirement for activation”. In my opinion it is a specific requirement because with Arg or Lys the protein would be “constitutively activated”. I think we agree on this, but the way it is written creates some confusion in my opinion. I suggest to change the sentence as follows “These results demonstrate that the protonation of histidine residues at positions 52 and 117 at acidic pH provides the molecular switch to ensure activation of CpxA below a given pH. Notably, the loss of responsiveness to acidification observed by replacing His with Lys or Arg residues indicated that a positive charge in these position is a key determinant and can lead to constitutive activation of *fabA*”.

Figure 5a,b – I don’t agree with the scale starting from a value which is not 0 (zero). OK for Fig. 5c (which is in log scale).

Page 15, line 292 – “Trail” should be “trial”.

Page 15, line 304 and 306 – somewhere here I suggest to state that the pH without adjustment to 7.0 was 5.2-5.5.

Page 20, line 408 – “decreases in pH protonate” should be “a decrease in pH protonates”

Reviewer #3 (Remarks to the Author):

The most questionable points are detailed below.

Major revision

Line 42: The “stationary phase alternative σ factor σ^S RpoS” phrase is redundant, change by “the alternative σ factor (RpoS)”

Line 77: Change by “adapts” and “grows”

Lines 113-114: The phrase “For a certain strain growing under certain conditions, the change of CFU ratio is caused by the change of growth at pH 4.2.” does not make sense, it must be deleted since it does not contribute anything new.

Lines 117-118: The authors should mentioned here that Phospholipids were extracted from exponential phase.

Fig 1d and all figures showing proteins levels: The authors must modify the Y axis legend to indicate that it is “relative amount of proteins” as was indicated for mRNA.

Lines 138-141: The authors must explain the difference observed between wild type strain and *fabBTs* mutants. The latter is considerably more acid-resistant at 30°C than the parental strain (Fig. 1g)

Line 149: Figure 2a is not necessary and should be removed, does not show results. However, the Supplementary Fig 2 should be placed as the main figure (Fig. 2a), since it is where the authors demonstrate the results mentioned in the text.

Line 161: Change the phrase "and cpxR knockout mutant displayed higher susceptibility to acidic challenge than wild-type strain" by "since cpxR knockout mutant displayed higher susceptibility to acidic challenge than wild-type strain even overexpressing nlpE"

Lines 168-172: Change the phrase by something like that: "As was previously reported, for fabA gene two transcription initiation sites were detected under growth at pH 7.0: S1 positively regulated by FadR30 and S2 negatively regulated by FabR31. Interestingly, when the cell growth at pH 4.2 our results indicated that the fabA gene has a third transcription initiation site located at the 203 bp upstream of the start codon, the S3 (summarized in Fig. 3a)."

Lines 177-178: Change the phrase by "indicating that they had no obvious effect on fabA and fabB gene expression under CpxRA-dependent activation (Supplementary Fig. 3)."

Lines 182-183: Change the phrase by "in which the CpxR binding box on fabA or fabB promoter region was replaced by CATCT-(5 nt)-CATCT sequence"

Lines 223: The authors must clarify whether it is a chromosomal or plasmid fusion construction.

Lines 234-236: Here the authors should have performed a positive induction control, using acetoacetate, to confirm that the construction is functional.

Lines 247-248: The authors must clarify whether they are chromosomal or plasmid mutations.

Lines 284-288: To compare the decrease or internal maintenance of pH, the authors should perform same assay but at pH7. In the way that the results were shown, it cannot be concluded if there is a decrease in the wild strain or that there is an increase in the strains overexpressing fabA or fabB. The control is necessary.

Line 293: Change "substituted fabA CpxR site" by "substituted fabA CpxR binding site" or by "fabA CpxR binding box mutant"

Line 298: Change parent for "parental"

Lines 306-308: According to the graph 's pH values, the 3HP production data was observed at approximately pH5 What happens when the medium is at pH 4.2, object of study of this work?

Line 314: Change "Salmonella typhimurium" by "Salmonella Typhimurium" here and throughout the text, figures and figure legends.

Line 324: The CpxR binding box does not seem to be much conserved in some of the cited species, what is the similarity percentage? Is has been experimentally proven that some of these sites could be occupied by CpxR? Add references if yes.

Lines 327-329: This sentence must be modified, since the ATR system does not have an exponential phase but acts or is activated during this phase.

Line 348 and throughout the discussion section: The authors should not repeat the results or mention the figures, they should discuss them.

Lines 348-349: This phrase is not understood, does not provide any information, or something is missing.

Lines 353-355: What is the result of this study that contribute to the mentioned topic?

Line 446: Change "CpxA from" by "CpxA under"

Materials and Methods:

The measure units must be unified throughout the materials and methods section, in accordance with the Journal 's rules (ul or uL; ml or mL, etc)

Lines 506: The sentence "and the same amount of target protein were separated in 12% SDS-PAGE" is incorrect, if load the same amount of "target" protein you will not be able to observe the differences represented in Fig 1d. This must be modified correctly.

Minor revision

Line 16: add "," after fluidity, and remove "and"

Line 18: remove "," after intestine

Line 31: Add "(E. coli)" abbreviation after Escherichia coli

Lines 58 to 60: Change "," by ";"

Line 100: Delete "," after washed

Line 119: Delete "an" after to such

Line 199: Add a "," after CpxA

Line 257: add "." after respectively; and start a new sentence using a connector like "We discover"

Line 304: put "previously" to the beginning of the sentence

Line 339: Change "susceptibility" by "sensitivity"

Line 351: Change "and achieve" by ", achieving"

Line 373: Delete "here"

Line 412: Change "ultimately leading" by "lead to"

Lines 429-431: Authors should improve punctuation marks.

Line 440: Change "All these" by "All of these data"

Ref: NCOMMS-19-18242A

An acid-tolerance response system protecting exponentially growing
Escherichia coli

Response to reviewers' comments:

Reviewer #1:

*1. lines 105/106 should read = (CFU after acid challenge/CFU before acid challenge)
x 100%*

Reply: It was revised as suggested (page 7, line 124-125).

2. line 121 should read "as a consequence"

Reply: It was revised as suggested (page 8, line 140).

*3. lines 209/210 would probably be better written as "where a different internal and
external pH can be stably maintained"*

Reply: This sentence was rephrased as suggested (page 12, line 228-229).

4. lines 372 change to "is likely to play a key role"

Reply: It was modified as suggested (page 19, line 388).

5. line 396 change to "mutans"

Reply: It was revised as suggested (page 20, line 16).

*6. line 406 needs rephrasing, I think to something like "will potentially enable the
synthesis of..."*

Reply: This sentence was rephrased as suggested (page 20, line 413).

Reviewer #2:

1. Page 1 , line 5 - Running Title: Replace “Acid Resistance” with “Acid Tolerance”

Reply: The running title was revised as suggested (page 2, line 25).

2. Page 2 – Abstract: two abbreviation (ATR and UFAs) are used, but reused only once. Maybe they could be removed in the abstract?

Reply: Following your suggestion, the abbreviation of UFAs was removed (page 3, line 36).

3. Page 2, line 9 – “Here, we characterized a new acid tolerance response (ATR) system and its regulatory circuit, required for *E. coli* growth at pH 4.2” should be replaced with “Here, we show for the first time the regulatory circuit controlling the acid tolerance response (ATR) system required for *E. coli* growth at pH 4.2” (see also the point immediately below)

Reply: As the words of “new”, “novel”, “for the first time” are not allowed by the journal, we rephrased this sentence as “we characterized an acid tolerance response (ATR) system and its regulatory circuit, required for *E. coli* exponential growth at pH 4.2.” (page 3, line 31-33)

4. Page 2, lines 18-19 – About the sentence starting with “This new ATR system is conserved.....” because the occurrence of this system is NOT new (Lin et al., 1995 reported it) I suggest to rephrase “The molecular aspects of this ATR system, which is conserved across bacteria species such as *Salmonella*, *Shigella*, *Klebsiella*, *Pseudomonas*, are herein dissected for the first time”

Reply: Same as the above, we can not use “for the first time”. Following the editor’s suggestion, this sentence was rephrased “Furthermore, this ATR system appears to be conserved in other Gram-negative bacteria.” (page 3, line 40-41)

5. Page 2, line 21 – replace the word “resistance” with “tolerance”

Reply: In the revised manuscript, this sentence has been removed.

6. Page 2, line 24 – Keywords: *don't think "Acid Resistance" is anymore necessary?*

Reply: The keyword "acid resistance" was removed. (page 3, line 43)

7. Page 2, line 26 – Keywords: *"membrane lipid" should be "membrane fluidity" or "membrane lipid composition"*

Reply: The keyword "membrane lipid" was revised to "membrane lipid composition" as suggested. (page 3, line 44-45)

8. Page 5, line 73 – *"nongrowth" should be "no-growth"?*

Reply: It was revised as suggested (page 6, line 92).

9. Page 7, line 121 – *After listing the different fatty acids I suggest to the authors to include the sentence that they used in the rebuttal to my comment "These five fatty acids represent more than 96% of total fatty acids in E. coli cells, consistent with those results reported previously (Int J Food Microbiol, 1997, 37: 163-173; Arch Microbiol, 1991, 157: 49-53.)"*

Reply: The sentence and associated references were added in the revised manuscript. (page 8, line 138-140)

10. Figure 2a – *Along the line representing the chromosome of E. coli BW25113 the slash symbols "/" should be correctly placed before fabA, between fabA and fabB and after fabB gene to clearly show that the drawing is not "in scale".*

Reply: Fig. 2a was modified as suggested.

11. Page 13, line 261 – *Remove "partial"*

Reply: It was revised as suggested (page 14, line 279).

12. Page 13, lines 261-264 – *Still not sure why the authors keep saying that the*

presence of the His residue I position 52 and 117 is “not a specific requirement for activation”. In my opinion it is a specific requirement because with Arg or Lys the protein would be “constitutively activated”. I think we agree on this, but the way it is written creates some confusion in my opinion. I suggest to change the sentence as follows “These results demonstrate that the protonation of histidine residues at positions 52 and 117 at acidic pH provides the molecular switch to ensure activation of CpxA below a given pH. Notably, the loss of responsiveness to acidification observed by replacing His with Lys or Arg residues indicated that a positive charge in these positions is a key determinant and can lead to constitutive activation of fabA”.

Reply: We insist that protonation of residues at positions 52 and 117 is required to maintain the response of CpxA to acidic pH, but the presence of a histidine at those positions is not a specific requirement for activation. In Fig. 4g, CpxA mutants (H52R, H52K, H117R, H117K) could activate the expression of FabA protein under pH 4.2, resulting in much higher level of FabA protein than the wild-type strain under pH 7.0. In Supplementary Fig. 5, CpxA mutants (H52R, H52K, H117R, H117K) presented similar level of FabA expression with the wild-type CpxA protein under pH 7.0. Therefore, these CpxA mutants are still response to the acidic pH, not activated constitutively. So, we think that the presence of histidine at positions 52 and 117 is not a specific requirement for CpxA activation under pH 4.2. (page 14, line 280-283)

13. Figure 5a,b – I don't agree with the scale starting from a value which is not 0 (zero). OK for Fig. 5c (which is in log scale).

Reply: Fig. 5a and 5b were revised with the scale starting from 0 in the revised manuscript.

14. Page 15, line 292 – “Trail” should be “trial”.

Reply: It was revised as suggested (page 16, line 311).

15. Page 15, line 304 and 306 – somewhere here I suggest to state that the pH without

adjustment to 7.0 was 5.2-5.5.

Reply: In the revised manuscript, we state that the pH without adjustment decreased to pH 5.2-5.5. The associated sentence was rephrased as “In shaking flask cultivation, the pH of culture decreased to pH 5.2-5.5 along with the production of 3HP, repressing the further production and cell growth (Fig. 5e). So, the pH of fermentation broth had to be adjusted to 7.0 periodically.” (page 16, line 321-323)

16. Page 20, line 408 – “decreases in pH protonate” should be “a decrease in pH protonates”

Reply: It was revised as suggested (page 21, line 425).

Reviewer #3:

1. Line 42: The “stationary phase alternative σ factor σ^S RpoS” phrase is redundant, change by “the alternative σ factor (RpoS)”

Reply: It was revised as suggested (page 4, line 59).

2. Line 77: Change by “adapts” and “grows”

Reply: It was revised as suggested (page 6, line 94).

3. Lines 113-114: The phrase “For a certain strain growing under certain conditions, the change of CFU ratio is caused by the change of growth at pH 4.2.” does not make sense, it must be deleted since it does not contribute anything new.

Reply: This sentence has been removed as suggested. (page 7, line 130)

4. Lines 117-118: The authors should mention here that Phospholipids were extracted from exponential phase.

Reply: We referred to that phospholipids were extracted from exponential phase cells in the revised manuscript (page 7, line 132-133).

5. *Fig 1d and all figures showing proteins levels: The authors must modify the Y axis legend to indicate that it is “relative amount of proteins” as was indicated for mRNA.*

Reply: The Y axis legend was modified to “relative amount of proteins” as suggested.

6. *Lines 138-141: The authors must explain the difference observed between wild type strain and fabBTs mutants. The latter is considerably more acid-resistant at 30°C than the parental strain (Fig. 1g)*

Reply: As you mentioned, the fabB_{Ts} mutant was moderately more tolerant to acidic pH than the wild-type strain at 30 °C. We don't know the reason yet, probably because the expression level of FabB could be changed slightly in these temperature sensitive mutants.

7. *Line 149: Figure 2a is not necessary and should be removed, does not show results. However, the Supplementary Fig 2 should be placed as the main figure (Fig. 2a), since it is where the authors demonstrate the results mentioned in the text*

Reply: Fig. 2a shows the genomic location of fab genes and corresponding CpxR binding sites, which is required by the Reviewer #2 in the first round of revision. As the alignment of conserved CpxR site sequence and fab gene promoters presented our result of sequence comparison, it was shown as Fig. 2b in the revised manuscript.

8. *Line 161: Change the phrase “and cpxR knockout mutant displayed higher susceptibility to acidic challenge than wild-type strain” by “since cpxR knockout mutant displayed higher susceptibility to acidic challenge than wild-type strain even overexpressing nlpE”*

Reply: This sentence was rephrased as suggested. (page 9, line 176-178)

9. *Lines 168-172: Change the phrase by something like that: “As was previously*

reported, for fabA gene two transcription initiation sites were detected under growth at pH 7.0: S1 positively regulated by FadR30 and S2 negatively regulated by FabR31. Interestingly, when the cell growth at pH 4.2 our results indicated that the fabA gene has a third transcription initiation site located at the 203 bp upstream of the start codon, the S3 (summarized in Fig. 3a)."

Reply: This sentence was rephrased as suggested. (page 10, line 183-188)

10. Lines 177-178: Change the phrase by "indicating that they had no obvious effect on fabA and fabB gene expression under CpxRA-dependent activation (Supplementary Fig. 3)."

Reply: It was revised as suggested. (page 10, line 193-194)

11. Lines 182-183: Change the phrase by "in which the CpxR binding box on fabA or fabB promoter region was replaced by CATCT-(5 nt)-CATCT sequence"

Reply: It was revised as suggested. (page 10, line 198-199)

12. Lines 223: The authors must clarify whether it is a chromosomal or plasmid fusion construction.

Reply: The gene encoding PhoQ-CpxA fusion protein locates on E. coli chromosome, and we added this information in the revised manuscript. (page 12, line 238-241)

13. Lines 234-236: Here the authors should have performed a positive induction control, using acetoacetate, to confirm that the construction is functional.

Reply: In the experiment shown in Fig. 4d and e, we verified the sensitivity of CpxA periplasmic domain to acidification, still sensing the acidic pH even in a PhoQ-CpxA fusion protein. Next, we are trying to prove that the periplasmic domain is necessary for CpxA to sense acidic environments, and carried out experiments shown in Supplementary Fig. 3. When its own periplasmic domain was replaced by that of AtoS, this CpxA mutant could not activate

expression of fab genes in vivo, and could not phosphorylate CpxR in vitro under pH 4.2. These results were sufficient to demonstrate the necessity of CpxA periplasmic domain, no matter whether the CpxA-AtoS fusion protein can sense acetoacetate.

14. Lines 247-248: The authors must clarify whether they are chromosomal or plasmid mutations.

Reply: The site-directed mutagenesis of cpxA gene was carried out on a recombinant plasmid pTrc-cpxA, and this information was added in the revised manuscript. (page 13, line 262-264)

15. Lines 284-288: To compare the decrease or internal maintenance of pH, the authors should perform same assay but at pH7. In the way that the results were shown, it cannot be concluded if there is a decrease in the wild strain or that there is an increase in the strains overexpressing fabA or fabB. The control is necessary.

Reply: When growing under pH 5.0-9.0, E. coli can maintain the intracellular pH between 7.4 to 7.9, which has been proved by many previous studies (Appl Environ Microbiol, 1990, 56: 1038-45; Proc Natl Acad Sci USA, 1981, 78: 6271-5). In this study, we measured the intracellular pH of E. coli cells growing under pH 4.2, and found the intracellular pH of wild-type cells decreased to pH 6.84, while overexpression of FabA or FabB kept the pH between 7.3-7.5. (page 15, line 301-302)

16. Line 293: Change “substituted fabA CpxR site” by “substituted fabA CpxR binding site” or by “fabA CpxR binding box mutant”

Reply: It was changed to “substituted fabA CpxR binding site”. (page 15, line 310)

17. Line 298: Change parent for “parental”

Reply: It was revised as suggested. (page 16, line 315)

18. Lines 306-308: *According to the graph's pH values, the 3HP production data was observed at approximately pH5 What happens when the medium is at pH 4.2, object of study of this work?*

Reply: In this experiment, minimal medium at pH 7.0 was used, and 3HP produced by recombinant E. coli strain acidified the fermentation broth to pH 5, which repressed the further production and cell growth. We didn't adjust the pH of broth during fermentation. In bioproduction of organic acids, product accumulation cause acidification of fermentation broth and inhibit bacteria growth at concentrations far below what is required for economical production. Now, large quantity of base titrant are required to raise pH of the medium in organic acid production process, and large amounts of acid must be consumed to recover the organic acids in the protonated form after production. Using an acid tolerant bacterial strain, the additional consumption of acid and base titrant will be circumvented and the overall production cost will be lowered remarkably.

19. Line 314: *Change "Salmonella typhimurium" by "Salmonella Typhimurium" here and throughout the text, figures and figure legends.*

Reply: It has been revised as suggested. (page 16, line 331; page 17, line 336; and other places throughout the manuscript)

20. Line 324: *The CpxR binding box does not seem to be much conserved in some of the cited species, what is the similarity percentage? Is has been experimentally proven that some of these sites could be occupied by CpxR? Add references if yes.*

Reply: The putative CpxR binding sites listed in Table 1 possess 6-8 bases same with the CpxR site consensus, which was added in Table 1 in the revised manuscript. The following figure is the sequence logo for the CpxR recognition weight matrix in E. coli (J Biol Chem, 2002, 277: 26652-61), indicating that some positions, especially in the second GTAAA repeat, are variable.

Furthermore, some experimentally identified CpxR binding sites also showed relative low identity with the CpxR site consensus, such as GTTACggaacTTTAC for rpoErseABC (J Biol Chem, 2002, 277: 26652-61) and GCAACtcctGAAAC for IdtD (J Bacteriol, 2015, 197: 603-14). The CpxR boxes in Table 1 are not experimentally proven except the E. coli ones identified in this study.

[Redacted]

21. Lines 327-329: This sentence must be modified, since the ATR system does not have an exponential phase but acts or is activated during this phase.

Reply: This sentence was modified to “All these facts suggest this ATR system functioning in exponential phase is highly conserved across bacteria species.” (page 17, line 345-346)

22. Line 348 and throughout the discussion section: The authors should not repeat the results or mention the figures, they should discuss them.

Reply: We have deleted some repeated description of results in the discussion section.

23. Lines 348-349: This phrase is not understood, does not provide any information, or something is missing.

Reply: This sentence has been rephrased to “Compared with those previously known AR systems, this growth-conferring ATR system is expected to have

more important physiological significance.” (page 18, line 363-365)

24. *Lines 353-355: What is the result of this study that contribute to the mentioned topic?*

Reply: It has been proved that de novo mutations play a critical role in the development of stress resistance in many bacterial species. Normally susceptible populations of bacteria may become resistant to environmental stress through mutation and selection. E. coli strain with activated UFAs-CpxRA system can exponentially grow under pH 4.2, and is expected to accumulate more mutations than the strain surviving but not growing under this condition. As a selection pressure, the acidic environments will enrich E. coli mutants with higher acid tolerance. Therefore, we speculate that this growth-conferring ATR system provides an opportunity for the evolution of novel AR and ATR systems. This is a hypothesis yet, and we will test it experimentally in the future.

25. *Line 446: Change “CpxA from” by “CpxA under”*

Reply: It has been revised as suggested. (page 22, line 460)

26. *Materials and Methods: The measure units must be unified throughout the materials and methods section, in accordance with the Journal’s rules (ul or uL; ml or mL, etc)*

Reply: The measure units were revised according to the journal’s instruction.

27. *Lines 506: The sentence “and the same amount of target protein were separated in 12% SDS-PAGE” is incorrect, if load the same amount of “target” protein you will not be able to observe the differences represented in Fig 1d. This must be modified correctly.*

Reply: This sentence was rephrased to “the same amount of protein sample were separated in 12% SDS-PAGE”. For Fig. 1d, the same amount of total

cellular proteins extracted from cells after 1 h of exposure to pH 7.0 and 4.2 were analyzed by western blot. (page 25, line 525)

28. *Line 16: add “,” after fluidity, and remove “and”*

Reply: It has been revised as suggested. (page 3, line 37)

29. *Line 18: remove “,” after intestine*

Reply: It has been revised as suggested. (page 3, line 39)

30. *Line 31: Add “(E. coli)” abbreviation after Escherichia coli*

Reply: It has been revised as suggested. (page 4, line 48)

31. *Lines 58 to 60: Change “,” by “;”*

Reply: It has been revised as suggested. (page 5, line 75-77)

32. *Line 100: Delete “,” after washed*

Reply: It has been revised as suggested. (page 7, line 117)

33. *Line 119: Delete “an” after to such*

Reply: It has been revised as suggested. (page 7, line 134)

34. *Line 199: Add a “,” after CpxA*

Reply: We don't think a comma is needed there. (page 11, line 216)

35. *Line 257: add “.” after respectively; and start a new sentence using a connector like “We discover”*

Reply: It has been revised as suggested. (page 14, line 274)

36. *Line 304: put “previously” to the beginning of the sentence*

Reply: It has been revised as suggested. (page 16, line 319-321)

37. Line 339: Change “susceptibility” by “sensitivity”

Reply: This sentence has been removed in the revised manuscript.

38. Line 351: Change “and achieve” by “, achieving”

Reply: It has been revised as suggested. (page 18, line 366-367)

39. Line 373: Delete “here”

Reply: It has been revised as suggested. (page 19, line 388)

40. Line 412: Change “ultimately leading” by “lead to”

Reply: It has been revised as suggested. (page 21, line 426)

41. Lines 429-431: Authors should improve punctuation marks.

Reply: This sentence was revised to “CpxRA upregulates some genes involved in cell wall modification, including peptidoglycan (PG) cross-linking proteins YcfS, YcbB and DacC; PG cleaving proteins AmiA, AmiC and Slr.” (page 21, line 441-443)

42. Line 440: Change “All these” by “All of these data”

Reply: It has been revised as suggested. (page 22, line 454)